# Visual control during climbing: Variability in practice fosters a proactive gaze pattern

Guillaume Hacques[ID]1*, Matt Dicks[ID]2, John Komar3, Ludovic Seifert[ID]1,4

1 Center for the Study and the Transformation of Physical Activities (CETAPS EA3832), Faculty of Sport Sciences, University of Rouen Normandy, UNIROUEN, Mont-Saint-Aignan, France, 2 School of Sport, Health and Exercise Science, University of Portsmouth, Portsmouth, United Kingdom, 3 National Institute of Education, Nanyang Technological University, Singapore, Singapore, 4 Institut Universitaire de France (IUF), Paris, France

* guillaume.hacques@univ-rouen.fr

**Data Availability Statement:** All relevant data are within the paper and its Supporting Information files.

**Funding:** This project received the support of the French National Agency of Research awarded to LS

## Abstract

In climbing, the visual system is confronted with a dual demand: controlling ongoing movement and searching for upcoming movement possibilities. The aims of the present research were: (i) to investigate the effect of different modes of practice on how learners deal with this dual demand; and (ii) to analyze the extent this effect may facilitate transfer of learning to a new climbing route. The effect of a constant practice, an imposed schedule of variations and a self-controlled schedule of variations on the gaze behaviors and the climbing fluency of novices were compared. Results showed that the constant practice group outperformed the imposed variability group on the training route and the three groups climbing fluency on the transfer route did not differ. Analyses of the gaze behaviors showed that the constant practice group used more online gaze control during the last session whereas the imposed variability group relied on a more proactive gaze control. This last gaze pattern was also used on the transfer route by the imposed variability group. Self-controlled variability group displayed more interindividual differences in gaze behaviors. These findings reflect that learning protocols induce different timing for gaze patterns that may differently facilitate adaptation to new climbing routes.

## Introduction

During the realization of complex everyday skill, gaze behaviors appear to smoothly support movement, but the acquisition of how learners acquire skill-relevant gaze patterns remains an under researched area [1, 2]. Complex situations, such as walking over rough terrain, commonly comprise a series of actions where the visual system is confronted with a dual-demand: (i) the accurate control of the current movement; and (ii) the anticipatory search for environmental demands that will constrain future movements [3, 4]. Studies have revealed a trade-off between the two demands that are modified to adapt to the immediate spatiotemporal conditions [5–7]. For example, when walking on different terrains, walkers adapt their gaze behavior to the difficulty of the surfaces [5]. This adaptation enables walkers to perform accurate foot placement on stable locations and to maintain an efficient locomotion pattern. However, it

(URL: https://anr.fr/Projet-ANR-17-CE38-0006, ID: ANR-17-CE38-0006 DynACEV). The funders had no role in study design, data collection and analysis, decision to publish, or preparation of the manuscript.

**Competing interests:** The authors have declared that no competing interests exist.

remains unclear whether different practice conditions can invite learners to acquire gaze patterns that are oriented towards one demand or the other. The current study aims to assess the effects of a constant and two variable practice conditions on the performance and gaze behaviors of learners in climbing to further understand how the visual control of action adapts to different practice conditions and how this may facilitate the transfer of learning.

## The timing of information pick-up and the dual demand on gaze control

When performing a continuous action in natural settings, the question of when and where to look in order to correctly control movement is of central importance. As information pick-up is dynamic, performers need to attend to the right information at the right time to guide skilled behavior [8]. The performer must find a balance between exploiting information for controlling their current movement and monitoring the environment for information that may constrain future movements [3, 5]. Walking across successive foot targets and walking over rough terrain represent paradigms that have been used to better understand how the visual system guides foot placement towards immediate and prospective targets [3, 6, 9, 10]. Research across these paradigms shows that the spatial demands of stepping accuracy impacts upon gaze behavior: the more accurate the performer's step needs to be, the more they use online control of their movements [5, 6]. The constraints are also temporal as the availability of visual information is necessary in critical phases of the step cycle to accurately perform foot placement and to maintain energetically efficient locomotion [10, 11]. Although dealing with this dual demand is necessary for everyone, it appears that skill level influences how individuals visually control their actions. For example, older adults who were identified as potential "fallers", responded to the dual-demand when approaching the target with a maladaptive gaze behavior that contributed to their poor stepping accuracy (and potential fall): their gaze shifted towards the next target before their heel had touched the ground in the immediate target [9]. In contrast, younger adults who were not at risk of a fall adopted a gaze pattern that resulted in them looking at the target until heel contact [9]. Thus, for precise visual control, it appears important to adopt a gaze pattern that prioritizes the accurate control of the current movement.

There is, however, evidence, which also points to conditions when gaze patterns are oriented towards future environmental conditions. For example, the visual control of movements when avoiding obstacles appears to change during development that coincide with the acquisition of different modes of locomotion [12]. For instance, infants that crawl or walk appear to rely more on the online visual guidance of their movements than children and adults who are able to proactively gaze toward obstacles (i.e., they can avoid obstacles while looking elsewhere) [12, 13]. Regarding the dual-demand, the proactive control of movements enables children and adults to anticipate what is coming ahead of them while infants–who are comparatively novices–needed to perform the actions one-by-one utilizing online control. Thus, the dual-demand on visual control can be affected by practice and coincides with the acquisition of motor skills.

In the realization of complex skills, performers are also often constrained by a dual-demand. For example, climbers need to control their ongoing movement while looking forward on the route to anticipate future movements. Climbing locomotion is performed on a vertical plane and climbers need to find support surfaces on the wall (the handholds and footholds) on which to apply forces to climb up the route while maintaining their balance with one to four points of contact (the hands and feet) [14]. In this context, the visual system is used (i) to locate the support surfaces on the wall, and (ii) to perceive and act on the opportunities for action that these supports and their configuration on the wall afford to the climber. Two studies have investigated the effect of practice on climbers' gaze behaviors [15, 16]. The first study

showed that after 6 trials, the number of fixations during ascents decreased without affecting the search rate (i.e., the number of fixations divided by the total duration of fixation) [15]. The second study investigated the changes in gaze behaviors of learners before and after performing 30 trials on the same climbing route (over 10 sessions). This study also assessed the transfer of learning by using three other routes that differed from the learning route by manipulating some properties of the handholds (i.e., the distances between handholds, their orientation or their shape) [16]. The results showed that the "quantity" of exploration (i.e., number of fixations) decreased and the gaze path–as measured by the entropy of the gaze transitions from hold to hold on the route—became less complex with practice. It appeared that gaze entropy was correlated with movement fluency, but only on the routes where the learners were attuned to the shape of the handholds. Taken together, these results suggest that with constant practice conditions, the demand on the anticipation of the future climbing movements decreased. However, the gaze entropy measure did not inform about the timing of the gaze movements relative to the climbing movements, which would reveal how performers deal with the dual-demand, either favoring online or proactive gaze control.

## Variability in the practice of perceptual-motor skill

According to learning approaches rooted in dynamical system theory, different practice schedules have the potential to differentially guide exploration, which could affect the transfer of learning [17]. These approaches have proposed three different forms of variability in practice: intrinsic variability, unstructured variability, and structured variability [18]. When the same practice condition is repeated, variability in the performed movement has been found to occur from one repetition to the next. This variability is intrinsic to the motor system. The second form of variability aims to provide additional random noise to the learners' movements during practice in order to find the global minimum of the perceptual-motor workspace and escape local minimums where individual intrinsic variability may be insufficient to facilitate learning [19]. This consists of adding unstructured variability to practice at the level of multiple task parameters [18]. Thus, it is hypothesized that unstructured variability in practice conditions may increase the learning rate of individuals and improve learning outcomes (i.e., retention and transfer) in comparison with constant practice conditions [19].

Unstructured variability may, however, be counterproductive if learners do not have the opportunity to stabilize the discovered movement patterns and optimize information-movement coupling [20]. A third form of practice variability motivated by the ecological approach to perception-action has revealed that the transfer of learning to different conditions occurs when learners attune to information during practice that is also available and reliable in transfer conditions [21]. In order to guide attunement, variability has been applied to practice conditions so that less useful information becomes unreliable during learning [22]. Learners in structured variable practice conditions have been found to attend to more reliable information resulting in better performance in a transfer task than learners in a constant practice group [21]. Moreover, the learners' attunement and ability to transfer learning differed according to the parameter of the task that was varied during practice [21, 23], which shows that although variable practice improves generalization, this generalization is specific to the learning conditions.

The rhythm of changes in learning conditions following structured variable practice are usually imposed upon the participants by the experimenters (e.g., [21]). However, studies have shown that even when learners are exposed to the same practice conditions, they demonstrate different learning dynamics (e.g., [24]). Thus, some participants may not benefit from variable practice if the externally imposed rate of exploration is too great for them to stabilize the newly

discovered movement solutions. A proposed solution is to give learners the opportunity to control when to change the practice conditions (e.g., [25–27]). For example, when participants controlled the difficulty of a rollerball task, they were shown to reach a success rate during practice that was better than participants experiencing practice with a progressive increase of difficulty [25]. Furthermore, participants that were given control of their practice schedule when practicing three sequences of a key-pressing task performed better on a transfer task than participants who had their practice schedules imposed by experimenters [27]. These results also showed that most of the participants chose to start practice using a blocked organization (i.e., with numerous repetitions of one of the tasks before switching to another one) before changing later in practice to become more variable. Thus, by giving control to the participants on the rate at which their learning conditions change, they appear to adapt the changes according to their skill level and their needs in terms of task exploitation. Therefore, learning outcomes in self-controlled practice appear to benefit from a ratio between exploration and exploitation during practice, which are sensitive to individual learning dynamics in comparison to constant practice or an imposed structured variable practice [25, 27].

In existing research, intrinsic variability and unstructured variability in practice have primarily been investigated using discrete multiarticular tasks (e.g., [28, 29]). In contrast, structured variable practice has been shown to improve the transfer of learning for a discrete anticipation task [23] and in continuous tasks where learners had to adapt their actions to the unfolding dynamics of the task [21, 22]. The perceptual-motor tasks used in these structured variable practice studies were performed in virtual environments to facilitate the control (and variations) of the available information. However, this poses questions concerning the transfer of the findings to natural settings. First, the possible movements of learners are restricted during virtual environments. Second, virtual environments may lead to the attunement to information, which may be detrimental for the transfer of learning from virtual environments to natural settings [30, 31]. In sum, the available information in natural settings is likely to be greater, and the movements usually involve more degrees of freedom, which gives learners an increased variety of opportunities to explore (i.e., to pick-up information) through their actions.

## The present experiment

It remains unclear how different practice conditions affect the acquisition of gaze behaviors and whether the changes in gaze behaviors are related to the learning outcomes. To address this gap, the aims of the current study were: (i) to investigate the effect of different types of practice on how learners deal with the dual-demand of gaze behavior; and (ii) to analyze the extent that this effect may be associated with the transfer of learning to a new climbing route.

In the current study, we compared the changes in performance and gaze behavior of three learning protocols on a training route, and we assessed the transfer of learning to a new route (i.e., the transfer route). We expected that, with practice, the learners in a constant practice condition (Constant group, CG) and the learners in a structured variable practice group (Imposed Variability Group, IVG) would differently balance the dual-demand of gaze behavior. We expected that the IVG would demonstrate more proactive gaze behaviors than the CG on both the training and transfer routes. These differences would enable the IVG to adopt a gaze behavior that is better adapted to climbing a new route as it would enable them to have a more proactive control of their climbing movements. Conversely, the gaze behavior developed by the CG on the training route may not be best adapted to climbing a new route as learning may be attuned to the training route, where continual exploration will likely reduce after extended practice in the same learning environment. Second, we examined whether giving the

participants the opportunity to control when to be confronted to a new climbing route (Self-controlled Variability group, SVG) would improve learning, and transfer of learning, in comparison to the group with an imposed schedule of climbing routes (the IVG). We expected that participants in SVG would benefit from learning to control the optimal ratio between exploration and exploitation during practice, which would result in better performance on the training and transfer routes. This optimal ratio would also translate into a gaze behavior that is less proactive than the IVG on both routes, suggesting a heightened skill in coupling information to movements.

## Method

### Participants

Twenty-four undergraduate students who volunteered to take part in the study were recruited (age: $M$ = 20.6 years, $SD$ = 1.1; 8 women and 16 men). Sample sizes were driven by the availability of participants (i.e., students able to attend to two learning sessions per week for 5 weeks) and previous work in this area (perceptual-motor learning in climbing; e.g., [15, 32, 33]). Their skill level was in the lower grade group according to the International Rock Climbing Research Association scale [34] as they had no or very little climbing experience. They all had normal or corrected to normal vision. Participants were randomly assigned to the CG, the IVG and the SVG. Before the first climbing session, the protocol was explained to all the participants, who then provided written informed consent to participate in this study. The protocol was approved by the French National Agency of Research (ID: ANR-17-CE38-0006 DynACEV) and conducted in accordance with the Declaration of Helsinki.

### Experimental design

**Learning protocols.** Participants attended 10 learning sessions that lasted for 5 weeks, with 2 sessions per week. The participants in the CG always climbed the same route, called the *training route* (Fig 1A, 84 trials in total). The participants in the IVG practiced on the *training route* (learning session 1) and on nine subsequent variations of the training route (the *variant routes*). Thus, the IVG practiced on a new variation of the training route in each session. The SVG followed the same protocol as the IVG with the difference that at the end of sessions 2 to 9, they were asked whether they wanted to continue practicing on the same route or if they wanted to change the route on which they performed the highest number trials. Thus, they could follow the same protocol as the IVG if they always chose to change the route. The content of the sessions is summarized in Table 1.

**Transfer test.** The transfer test consisted of two trials on a climbing route called *Transfer route*. The first trial was performed at the beginning of the first learning session and was used as a baseline. The second trial was performed after the last trial of the last learning session to examine the effect of the three learning protocols on the transfer of learning.

**Route design.** The experiment took place in a climbing gym where two walls were used: the first was used for the training route and the second for the transfer route and the variants. Two routes could be placed on the second wall. Each route was hidden with a tarpaulin so that participants could only see the route to be climbed. All routes were designed with the same two models of climbing holds (Fig 1B, Volx Holds®, Chessy-les-mines, France): one for hand-holds and one for footholds. The variants were designed with the same number of holds as the training route (i.e., 13 handholds and 7 footholds) and the transfer route was composed of 13 handholds and 6 footholds, but the layout of the holds on the wall differed between routes. The training route was 5.25 m high, and the other routes were 4.80 m high.

**A**

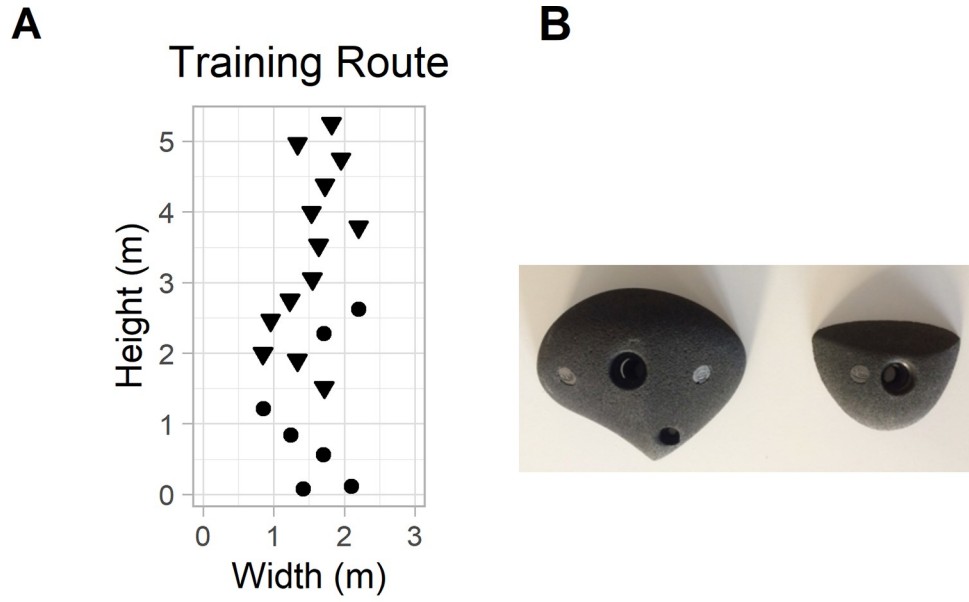

**B**

**Fig 1. Presentation of the training route and the climbing holds.** (A) Design of the training route. (B) Model of handhold (on the left) and model of foothold (on the right) used to create the routes.

**Instruction.** Prior to each trial during the learning sessions, the participants were provided with the following guidance: (i) to climb as fluently as possible, avoiding pauses and jerky movements of the body; (ii) to use all the handholds in an order from the bottom to the top of the wall; and (iii) to use all the handholds and footholds with a single limb contact at a time (participants couldn't use a hold with both hands or feet at once). These instructions were repeated before each trial to ensure that participants were aware that the goal of each climb

**Table 1. Program of the learning sessions for the three groups.**

|  | **Constant Group** | **Imposed Variability Group** | **Self-controlled Variability Group** |
|---|---|---|---|
| Session 1 | **1xTransfer \| 3xTR \| 3xTR** | **1xTransfer \| 3xTR \| 3xV1** | **1xTransfer \| 3xTR \| 3xV1** |
| Session 2 | 9xTR | 3xTR \| 3xV1 \| 3xV2 | 3xTR \| 3xV1 \| 3xV2 |
| Session 3 | 9xTR | 3xTR \| 3xV2 \| 3xV3 | 3xTR \| 3xV? \| 3xV? |
| Session 4 | 9xTR | 3xTR \| 3xV3 \| 3xV4 | 3xTR \| 3xV? \| 3xV? |
| Session 5 | 9xTR | 3xTR \| 3xV4 \| 3xV5 | 3xTR \| 3xV? \| 3xV? |
| Session 6 | 9xTR | 3xTR \| 3xV5 \| 3xV6 | 3xTR \| 3xV? \| 3xV? |
| Session 7 | 9xTR | 3xTR \| 3xV6 \| 3xV7 | 3xTR \| 3xV? \| 3xV? |
| Session 8 | 9xTR | 3xTR \| 3xV7 \| 3xV8 | 3xTR \| 3xV? \| 3xV? |
| Session 9 | 9xTR | 3xTR \| 3xV8 \| 3xV9 | 3xTR \| 3xV? \| 3xV? |
| Session 10 | **3xTR \| 3xTR \| 1xTransfer** | **3xTR \| 3xV9 \| 1xTransfer** | **3xTR \| 3xV? \| 1xTransfer** |

The table presents the content of each learning session for the three practice conditions. On the first and last session, the participants climbed a transfer route (Transfer). The participants in the Constant group climbed a training route (TR) 6 to 9 times per session. The participants in the Imposed Variability group climbed the training route 3 times on all the sessions and they climbed 9 variants routes (V1 to V9) across the learning protocol. The Self-controlled Variability group followed a similar protocol as the Imposed Variability group, but the number of variants routes discovered depended on the individuals choice during the practice. The data collected from the trials written in bold characters are those analyzed in the current study.

was to efficiently chain movements to reach the top of the climbing route, a climbing skill commonly referred to as route-finding [35].

**Procedure.** Each session lasted approximately 1 h and therefore, the entire study comprised a total of 240 hours data collection (testing and practice of all participants). Each session started with a 10 min warm-up in a bouldering area. The participant was equipped with climbing shoes, a harness and the mobile eye-tracker and was told the instructions. On the first session, one of the experimenters demonstrated how to climb in a bouldering area in accordance with the instructional prompts and invited the participants to try. Then, the participant warmed-up while familiarizing with the prompts in the bouldering area.

Then, the same procedure was performed for each trial: (i) the route to be climbed was uncovered, the others were hidden with a tarpaulin, (ii) the mobile eye tracker was calibrated, and the recording started, (iii) the participant stood 3m in front of the route for 30s of route preview. The participant could stop the preview when they wanted. During the preview, the experimenters started the video recording. (iv) The participants were top roped, that is, the rope was anchored at the top of the wall and to the participant for security during the ascents. (v) The prompts were provided by the experimenter to the participant. (vi) The experimenter then performed the synchronization procedure (see Synchronization Procedure). (vii) The participants were placed in the starting position, holding the first handhold with two hands and their feet were on the first two footholds. (viii) When the participants were ready and secured, the experimenter announced that they could start the climb. The climbed ended when the participants grasped the last handholds and remained immobile for a few seconds. (ix) The participant was then lowered down, and all the recordings were stopped.

## Data collection

**Contact time with holds.** The climbing walls were equipped with the Luxov Touch ® system (http://www.luxov-connect.com/en/products/#touch, Arnas, France) as used by [36]. This system uses a sensor technology to provide a measure of the time of contact and release of the handholds and footholds. The reported accuracy of the system is 1.57 ms at 99.7% confidence interval (see patent details: FR3066398-2018-11-23 / WO2018/211062A1-2018-11-22; https://patents.google.com/patent/WO2018211062A1/en). The starting time and ending time of the climb were obtained with this system. The start time was considered when the participant touched another hold other than the starting holds and the end of the climb was considered when the climber touched the last handhold. The time of the first contact with each of the handholds of the routes were also collected.

**Tracking of the hip trajectory.** Trials were filmed at 29.97 fps on 1920x1080 pixels frames with two GoPro 5 cameras (GoPro Inc. ®, San Mateo, CA, USA), each camera captured an entire wall. The cameras were placed at a height of 2.80m. On the back of the participants' harness, a light was placed.

The videos of the cameras were imported in Kinovea© (version 0.8.25, Boston, MA, USA). The lens distortion was corrected by importing the intrinsic parameters of the cameras lens in Kinovea from Agisoft lens (version 0.4.1, Agisoft LLC, Saint Petersburg, Russia). A manually set grid was used to correct the perspective and to calibrate the distances by using markers placed on the climbing routes. The light on the back of the participant was tracked from the reference frame (when the experimenter tapped the hold) until the moment the climber touched the last handhold of the route. The tracking was used to get the projected coordinates of the hip position on the 2D wall for each frame of the video. The starting and ending times obtained from the Luxov Touch system were used to cut the temporal series of the hip position to have the fluency measure corresponding to the climbing period.

**Gaze behavior.** The climber wore mobile eye-tracking glasses (Tobii Pro Glasses 2©, TobiiAB®, Sweden) on each trial. The glasses tracked the eye movements at a frequency of 50Hz with two cameras under each eye which after calibration, provided the gaze location on the video scene camera that recorded at 25fps on 1920x1080 pixels frames. Before each trial, the calibration was conducted by placing a target (diameter of the target is 43mm) at an arm's length from the standing participant. After the calibration, the accuracy of the gaze location was then checked by asking the participant to look at the target again. If the calibration failed or the point of gaze did not overlap with the target, the procedure was repeated until that the gaze location met the center of the target.

**Synchronization procedure.** The data from the mobile eye-tracker and the Luxov Touch system were synchronized by asking the participant to look at one hold while the experimenter tapped the location. Then the time of the first frame in the video of the eye tracker that showed the contact of the experimenter's finger with the hold was used as a reference time to synchronize the two. This synchronization was used to obtain the gaze offset time (see in the subsection Gaze Behavior within the section Data Analysis for more details and reliability measures regarding this dependent variable).

## Data analysis

**Climbing fluency.** The coordinates of the hip trajectory were used to compute the geometric index of entropy (GIE). The GIE was designed as a measure of performance that reflects the degree of coherence in information-movement couplings [37]. The GIE enables assessment of the degree of complexity of the hip trajectory. A complex hip trajectory would reflect a poor sensitivity of the climber to the environmental constraints, whereas a smooth trajectory would reflect fluent climbing movements. GIE is calculated with the following equation:

$$GIE = \log_2(\frac{2L}{c})$$

L is the length of the hip trajectory and c the perimeter of the convex hull around the hip trajectory. Data analyses to obtain the GIE values were performed with Matlab R2014a ® software (version 8.3.0.532, The MathWorks Inc., Natick, MA, USA).

**Gaze behavior.** Gaze behavior measures focused on the gaze patterns oriented towards the handholds and hand movements of the participants. The analyses of the mobile eye tracker recordings were performed using Tobii Pro Lab© (version 1.102.16417, TobiiAB, Sweden). The raw filter was applied to provide data on the location of the gaze position on each frame. A circle with a 20 cm radius around each of the 13 handholds was considered as areas of interest (AOI). Two different aspects of gaze behavior were coded for each ascent: (i) the last period the participant's gaze stayed within an AOI before touching the corresponding handhold for the first time in the trial, and (ii) the temporal series of the AOI locations that the point of gaze passed through.

For the first measure, the coder recorded the last period that the participant's point of gaze stayed within an AOI before touching the corresponding handhold for the first time in the trial [9]. This was repeated for each handhold of the route with the exception of the starting and last handhold ($N = 11$). Subsequently, the onset and offset time of each gaze period that related to these periods of gaze within an AOI before contact were recorded. If the onset or offset time could not be collected due to missing gaze samples, the entire period was not considered for analysis. The gaze onset and offset times were related to the contact time of the handhold given by the Luxov Touch system. Thus, the visits with a negative offset time would correspond to a proactive control of the hand movement, as the participant's gaze would have

move away from the AOI before the moment of contact with the handhold. Using the offset time, we calculated the proportion of online visits, that is, the proportion of visits with a positive offset time, meaning that participant's gaze was within the AOI at the moment of contact with the handhold. The duration of the gaze visit was also obtained from onset and offset time.

For the second measure, to be considered in the temporal series, the point of gaze needed to remain within the AOI of the handhold for more than 3 frames (i.e., 60ms), otherwise, it would be considered as an eye movement passing by the AOI and thus would not be coded [38]. Furthermore, the handhold would either be the one that was currently grasped by the participant, or another one above it. This coding procedure ensured that the results only informed about the gaze displacements relating to the current or future hand movements of the performer. The temporal series of visited AOIs was used to calculate the conditional visual entropy (H) measure, with the following equation [39]:

$$H = -\sum_{i=1}^{n} p(i) \left[ \sum_{j=1}^{n} p(i,j) \log_2 p(i,j) \right], i \neq j$$

with p(i) the probability of visiting the AOI i, and p(i,j), the probability that the point of gaze would shift from i to j. The higher the value of the conditional visual entropy, the more the gaze path went from one AOI to another in a random manner whereas a low value would reflect a structured gaze path [40]. It should be noted that, if the participants gaze shifted from one AOI to the next on the ascent, the value of H would be 0 as all p(i,j) would be 1. We expected that with practice, participants conditional visual entropy would tend to 0.

The reliability of the coding method was assessed on eight trials taken randomly. This sample was coded a second time by the original coder two months after the first coding and by a second researcher. For the three dependent variables relating to gaze behaviors, we performed Pearson correlations that showed that the intra-coder reliability ranged between $r = .994$ and $r = .997$ and the inter-coder reliability between $r = .991$ and $r = .993$. For the measure of the gaze offset time (which could be affected by the synchronization procedure), the intra-coder mean difference between the first and second coding was -0.9ms (*Mean 95% CI = [-4.1ms, 2.3ms]*, *SD = 14.6ms*) and the inter-coders mean differences was -0.2ms (*Mean 95% CI = [-3.8ms, 3.4ms]*, *SD = 16.3ms*).

*Global observations regarding the gaze sample.* The gaze behavior of one participant in the IVG was excluded from the analysis due to the loss of gaze data during the climbs. For the rest of the participants, the offset time and duration of the period of gaze within AOI was obtained for 91.3% of the visits on the training route and for 86.6% of the visits on the transfer route. There was no significant difference in the proportion of excluded periods in the three learning groups on the training route [$\chi2(2, N = 1205) = 0.98, p = .612$] and on the transfer route [$\chi2(2, N = 381) = 0.66, p = .719$].

## Statistical analysis

The dependent variables were submitted to separate mixed ANOVA with Session (2) as a within participant factor and Group (3) as a between participant factor. The Levene tests for homogeneity of variance and Shapiro-Wilk tests for normal distribution were performed before running the mixed ANOVA. If the tests were significant for GIE or conditional visual entropy, outliers were (i) identified with the identify_outliers() function from the rstatix package [41, 42], and (ii) replaced by the mean of the corresponding series and the tests were performed a second time. For the offset times and durations of the gaze visits, if the tests were significant, outliers were removed, and the tests were performed a second time. In the event of nonsignificant results in the mixed ANOVA, we also performed Bayesian mixed ANOVA and

reported the Bayes factors (*BF*) to assess the evidence in favor of the null or the alternative hypothesis [43, 44]. The *BF* values are interpreted according to the classification and thresholds presented by [45].The mixed ANOVA was followed by post-hoc tests with a Bonferroni correction of the *p*-value to examine the main factors Session and Group. In case of significant result regarding the interaction between Session and Group, planned contrast tests were used to examine the practice effect for each group, and to assess whether this practice effect was different between groups. The generalized eta squared ($\eta_G^2$) is reported as a measure of effect size with values of .02 as small, .13 as medium and .26 as large effect [46]. These statistical analyses were run using JASP [47].

## Results

Two participants only attended the first session and then dropped out and one participant injured herself after the fourth training session and could not continue the protocol. Unfortunately, we did not have the resources (time and availability of the climbing wall) to replace the three participants that dropped out, which equated to 60h of data collection (5 weeks of data collection for one participant represented approximately 20h of time). Thus, these three participants were not included in the statistical analyses and the final sample size was 21 participants (age: *M* = 20.6 years, *SD* = 1.1; 7 women and 14 men). The data used in the statistical analyses can be found in S1 Data.

### Practice schedule of the SVG

The self-controlled scheduling of the practice condition for participants in the SVG are displayed in the Table 2. All the participants chose at least once to practice on the same climbing route in the following session, thus none of the participants of the SVG followed the same practice schedule as the IVG. Although the participants were, in general, likely to vary the practice route, the proportion of participants who chose to keep the same route increased on the two last sessions, with the proportion of change decreasing to .50 in the last session.

### Changes in climbing fluency and gaze behaviors on the training route

**Climbing fluency.**   Fig 2 displays the GIE scores of the participants. The mixed ANOVA applied to the GIE showed a large effect of the factor Sessions [$F(1,18) = 275.73$, $p < .001$, $\eta_G^2$

**Table 2. Practice schedule of the participants in the SVG.**

| Participant | Session | | | | | | | |
|---|---|---|---|---|---|---|---|---|
| | 2 | 3 | 4 | 5 | 6 | 7 | 8 | 9 |
| P1 | | | █ | | | | █ | |
| P2 | | | | | | █ | | |
| P3 | | | | | | | | █ |
| P4 | | | | | | | | █ |
| P5 | | █ | | █ | █ | | | █ |
| P6 | | | | | █ | █ | █ | █ |
| P7 | █ | | | | | | | |
| P8 | | | █ | | | | █ | |
| Proportion of change | .875 | .875 | .750 | .875 | .750 | .750 | .625 | .500 |

Grey frames show when participants chose to maintain the same route on the following session, whereas white frames display when they chose to practice on a new route. The proportion of change reflects the proportion of participants who chose to practice a new route on the following session.

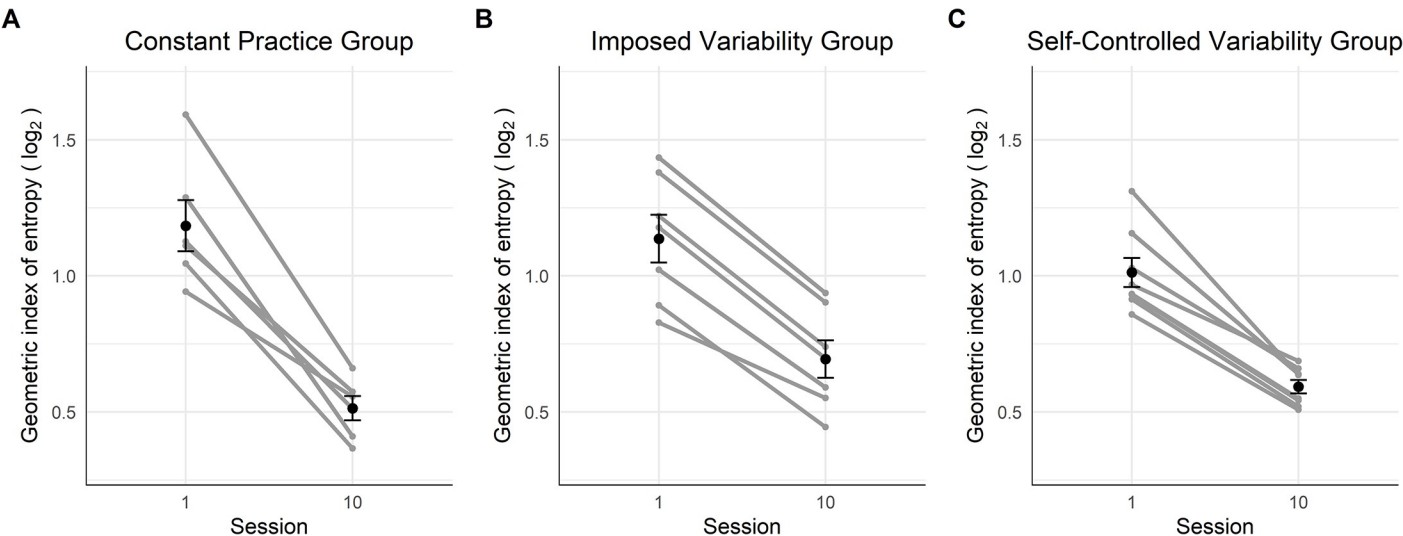

**Fig 2. Dynamics of the climbing fluency on the first and last session of the protocol for the three groups.** The black points represent the sessions mean and the error bars their standard error. The grey points and lines represent each participant's dynamics.

= .72], and a small effect of the interaction between the main factors of Session and Group [$F_{(2,18)}$ = 6.38, $p$ = .008, $\eta_G^2$ = .11], whereas the Group effect was not significant [$F_{(2,18)}$ = 1.00, $p$ = .389, $\eta_G^2$ = .08]. The results of the Bayesian mixed ANOVA suggested anecdotal evidence in favor of a Group effect ($BF$ = 2.17). The contrast tests revealed that participants across all three groups had more complex hip trajectories in session 1 than session 10 ($M$ = -0.51, $CI$ = [-0.58, -0.45], $ps$ < .001). This change in the spatial fluency score with practice was significantly higher for CG than for IVG ($M$ = -0.11, $CI$ = [-0.20, -0.03], $p$ = .009) but no significant difference was observed between the IVG and the SVG ($M$ = -0.01, $CI$ = [-0.09, 0.07], $p$ = .762).

**Complexity of the gaze path.** Fig 3 displays the visual entropy scores of the participants. Regarding the measure of the complexity of the gaze path (Fig 3), we performed the mixed

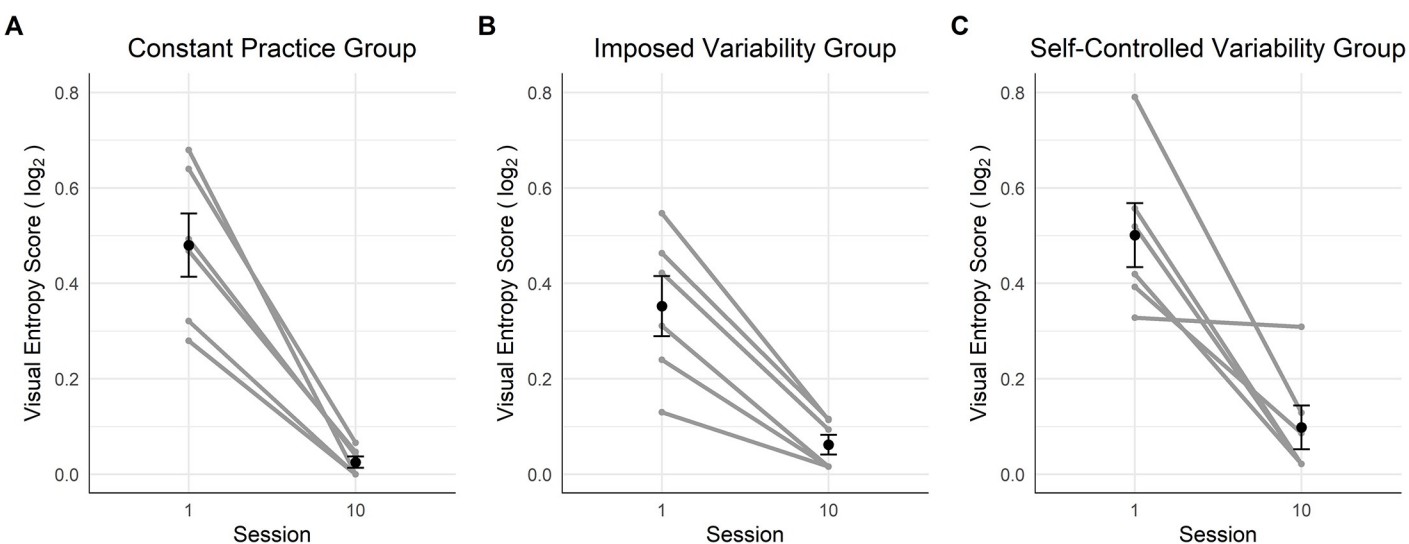

**Fig 3. Dynamics of the visual entropy score on the first and last session for the three groups.** The black points represent the sessions mean and the error bars their standard error. The grey points and lines represent each participant's dynamics.

ANOVA although the data on the session 10 were not normally distributed due to repeated values in the CG ($n = 3$). The mixed ANOVA showed a large effect of the factor Session [$F(1,15) = 93.06$, $p < .001$, $\eta_G^2 = .74$] but no significant effect of the factor Group [$F(2,15) = 1.52$, $p = .250$, $\eta_G^2 = .10$] and the interaction between Session and Group [$F(2,15) = 1.50$, $p = .255$, $\eta_G^2 = .08$]. The Bayesian mixed ANOVA suggested anecdotal evidence in favor of the null hypothesis for the factor Group ($BF = 0.58$), and anecdotal evidence in favor of an effect of the interaction between Session and Group ($BF = 1.03$). The post-hoc test revealed that participants' gaze showed less variability in session 10 compared to session 1 ($M = -0.38$, $CI = [-0.47, -0.30]$, $p < .001$).

**Characteristics of the last gaze visit.** *Offset time.* Fig 4 displays the offset times of the last gaze visits on handholds before touching them. The results of the mixed ANOVA showed a medium effect of the interaction between Session and Group on the offset time [$F(2,17) = 7.14$, $p = .006$, $\eta_G^2 = .16$], whereas the main factor Session [$F(1,17) = 0.18$, $p = .68$, $\eta_G^2 = .00$] and Group [$F(2,17) = 2.37$, $p = .124$, $\eta_G^2 = .18$] was not significant. The Bayesian mixed ANOVA suggests anecdotal evidences in favor of an effect of the main factors Session ($BF = 1.35$) and Group ($BF = 2.29$).

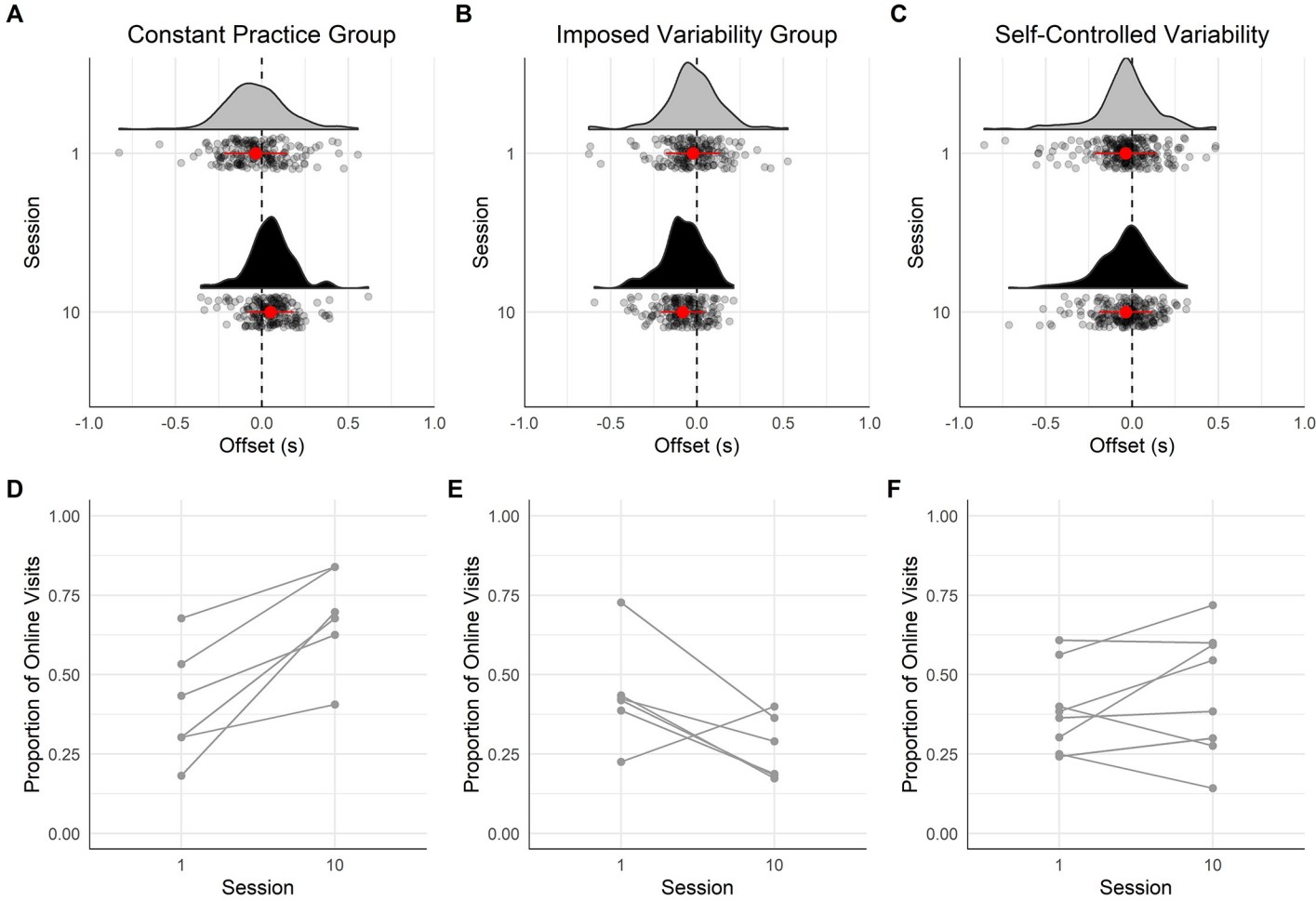

**Fig 4. Offset time of the last gaze visit before the hand contacted the handhold for the three groups on the training route.** In (A), (B) and (C), the vertical dashed line shows the time the hand touched the handhold, each point represents one gaze visit, the half violin shows the density of points, the red/grey point with the error bar refers the mean of all the gaze visits and the standard deviation around the mean. The color of the half violin refers to the learning session: in grey, session 1 and in black, session 10. (D), (E) and (F) displays the individuals' proportion of online visits on session 1 and 10. (A) and (D) show data for the constant practice group, (B) and (E) for the imposed variability group, and (C) and (F) for the self-controlled variability group.

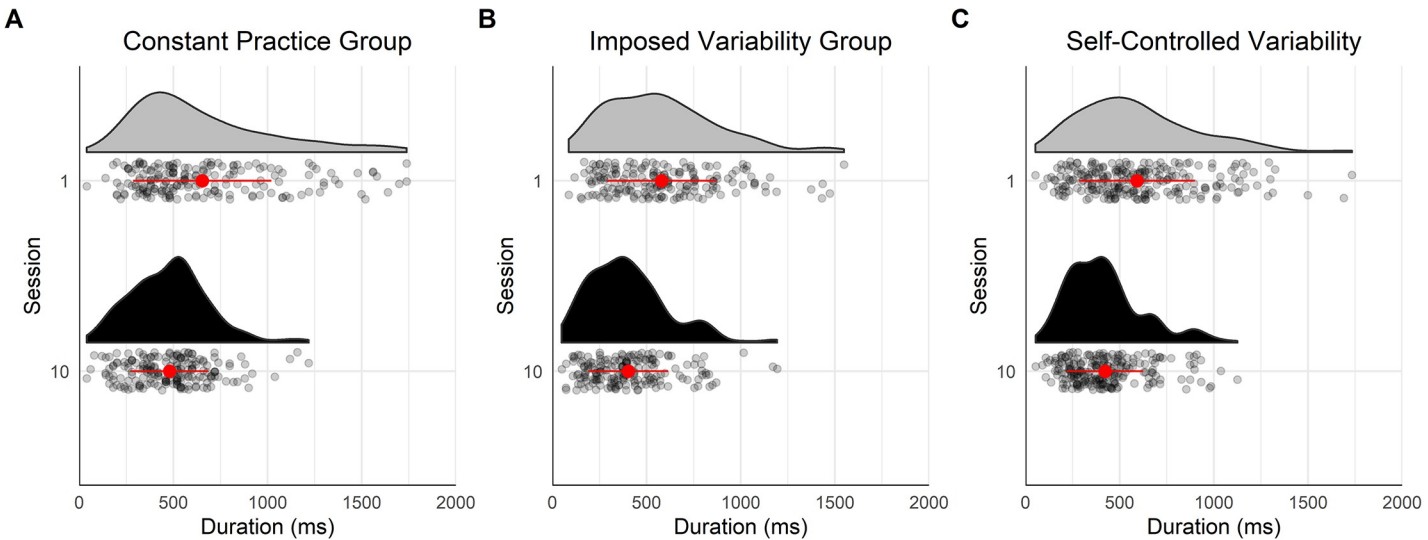

**Fig 5. Duration of the last gaze visit before the hand contacted the handhold for the three groups on the training route.** In (A), (B) and (C), each point represents one gaze visit before a contact with a handhold, the half violin shows the density of points, the red/grey point with the error bar refers to the mean of all the gaze visits and the standard deviation around the mean. The color of the half violin refers to the learning session: in grey, session 1 and in black, session 10. (A), (B) and (C) shows data for the constant practice group, imposed variability group and the self-controlled variability group respectively.

The contrast tests showed that the change in the visit offset time was different between CG and IVG with practice ($M$ = +68ms, $CI$ = [+30 ms, +106 ms], $p$ = .001) as the CG visit offset time occurred later on session 10 than on session 1 ($M$ = +74 ms, $CI$ = [+20 ms, +128 ms], $p$ = .010), whereas practice had the opposite effect on IVG as the visit offset time occurred earlier on session 10 than on session 1 ($M$ = -62 ms, $CI$ = [-116 ms, -8 ms], $p$ = .026). The change in the visit offset time with practice was not significantly different between IVG and SVG ($M$ = -34 ms, $CI$ = [-70 ms, +2 ms], $p$ = .060), whilst practice did not significantly affect the visit off-set time of SVG ($M$ = +6 ms, $CI$ = [-41 ms, +52 ms], $p$ = .798).

For the CG, the proportion of online visits increased between session 1 and 10, from .40 to .68 [$\chi^2$ (1, $N$ = 380) = 29.75, $p$ < .001] (Fig 4D). Conversely, the proportion of online visits decreased between the two sessions for the IVG, from .44 to .27 [$\chi^2$ (1, $N$ = 364) = 11.54, $p$ < .001] (Fig 4E). For the SVG, the proportion of online visits did not change significantly [.41, $\chi^2$ (1, $N$ = 461) = 1.87, $p$ = .171] (Fig 4F). Thus, while the CG appeared to favor online control of hand movements with practice, the IVG tended to adopt a proactive control of hand movements.

*Gaze duration*. Fig 5 displays the duration of the last gaze visit on handholds before touching them. The mixed ANOVA revealed a large effect of the main factor Session [$F(1,17)$ = 46.50, $p$ < .001, $\eta_G^2$ = .49] and no significant effect for the main factor Group [$F(2,17)$ = 0.99, $p$ = .394, $\eta_G^2$ = .07] and the interaction between Session and Group [$F(2,17)$ = 1.54, $p$ = .242, $\eta_G^2$ = .06]. The Bayesian mixed ANOVA suggested anecdotal evidence for the null hypothesis regarding the main factor Group ($BF$ = 0.50) and the interaction between Session and Group ($BF$ = 0.81). The post-hoc test showed that the duration of the visit was significantly shorter in session 10 in comparison to session 1 ($M$ = -122 ms, $CI$ = [-160 ms, -85 ms]), $p$ < .001).

## Changes in climbing fluency and gaze behaviors on the transfer route

**Climbing fluency.** The Levene test showed that the assumption of equality of variances was violated on session 1 [$F(2,18)$ = 6.36, $p$ = .008]. The mixed ANOVA applied to the GIE

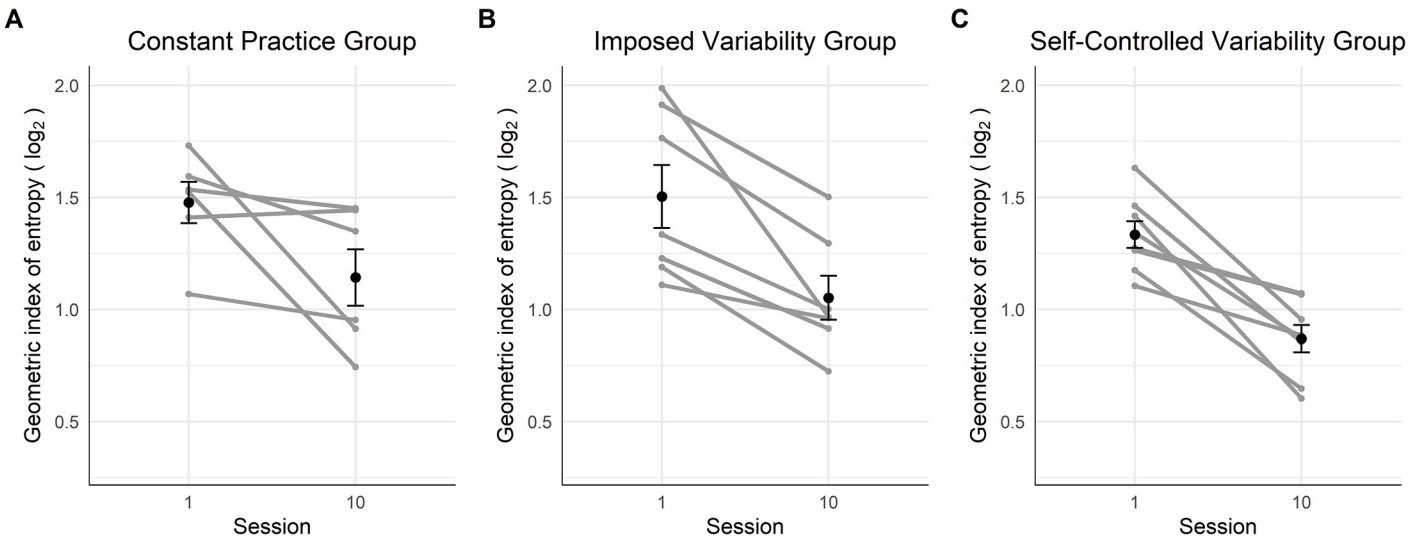

**Fig 6. Changes in the climbing fluency of the three groups on the transfer route.** The black points represent the sessions mean and the error bars their standard error. The grey points and lines represent each participant's dynamics.

revealed a large effect of the main factor Session [$F(1,18) = 42.38, p < .001, \eta_G^2 = .43$] but no significant effect for the factor Group [$F(1,18) = 2.06, p = .157, \eta_G^2 = .13$] and the interaction between Session and Group [$F(2,18) = 0.39, p = .685, \eta_G^2 = .01$]. The Bayesian mixed ANOVA suggested anecdotal evidence for the null hypothesis regarding the main factor Group ($BF = 0.77$) and the interaction between Session and Group ($BF = 0.62$). The post-hoc test showed that the hip trajectory of the participants was significantly less complex in session 10 in comparison to session 1 on the transfer route ($M = -0.42, CI = [-0.55, -0.28]$), $p < .001$) (Fig 6).

**Complexity of the gaze path.**   The mixed ANOVA applied to the visual entropy scores revealed a large effect of the main factor Session [$F(1,15) = 58.35, p < .001, \eta_p^2 = .60$] whereas the main factor Group [$F(2,15) = 0.22, p = .809, \eta_p^2 = .02$] and the interaction between Session and Group [$F(2,15) = 0.74, p = .495, \eta_G^2 = .04$] were not significant. The mixed Bayesian ANOVA suggested medium and anecdotal evidence in favor of the null hypothesis for the factor Group ($BF = 0.30$) and the interaction between Session and Group ($BF = 0.44$) respectively. The contrast test showed that the variability of the gaze path decreased on session 10 ($M = -0.40, CI = [-0.51, -0.29], p < .001$) (Fig 7).

**Characteristics of the last gaze visit.**   *Offset time.* Fig 8 displays the offset time of the last gaze visits on handholds before touching them. The results of the mixed ANOVA showed a medium effect of the factor Session [$F(1,17) = 16.88, p < .001, \eta_p^2 = .20$]. The factor Group [$F(2,17) = 0.73, p = .498, \eta_p^2 = .06$] and the interaction between Session and Group [$F(2,17) = 0.77, p = .479, \eta_p^2 = .02$] were not significant. The Bayesian mixed ANOVA suggested anecdotal evidence in favor of the null hypothesis for the factor Group ($BF = 0.43$) and the interaction between Session and Group ($BF = 0.52$). The post-hoc test revealed that the visit offset time occurred earlier in session 10 comparing to session 1 ($M = -72$ ms, $CI = [-109$ ms, $-35$ ms$], p < .001$).

The proportion of online visits was not significantly different between the three groups on session 1 of the transfer route (.53) [$\chi^2(2, N = 191) = 0.964, p = .618$] but was on session 10 [$\chi^2(2, N = 190) = 11.87, p = .003$]. More specifically, it appears that the CG maintained the proportion of online gaze visit on session 10 as with session 1 [.57, $\chi^2(1, N = 118) = 0.31, p = .577$] (Fig 8D). However, the IVG group utilized significantly less online visits in session 10 (.29) than in session 1 (.48) [$\chi^2(1, N = 121) = 4.98, p = .026$] (Fig 8E). The SVG maintained the same

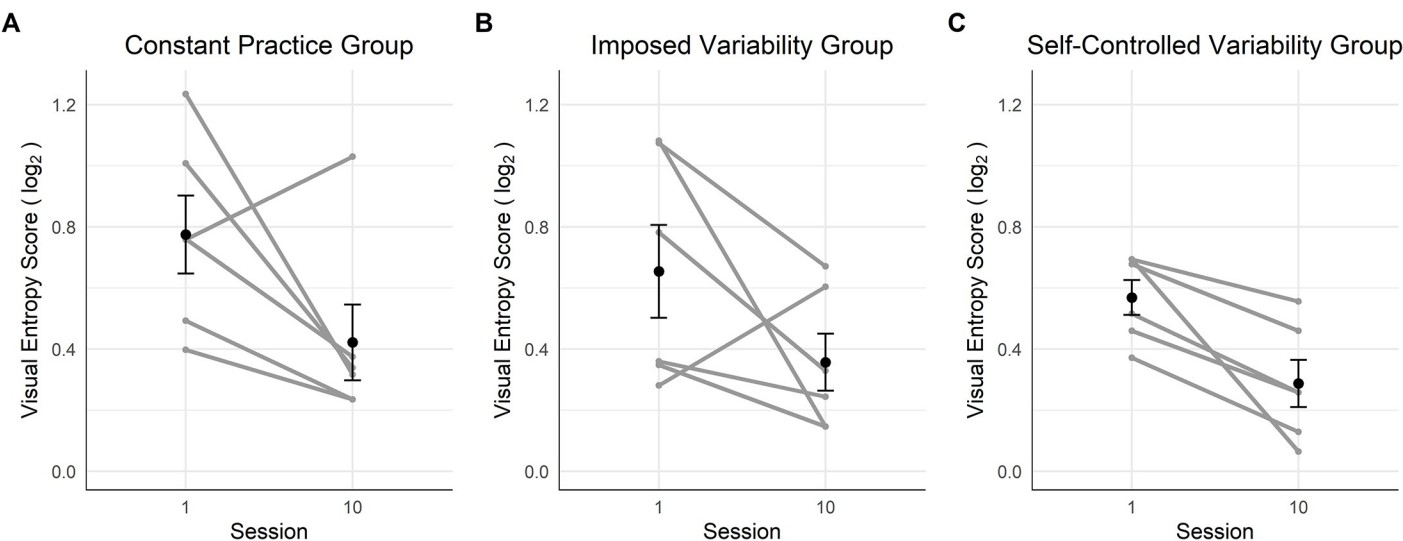

**Fig 7. Changes in the complexity of the gaze path for the three groups on the transfer route.** The black points represent the sessions mean and the error bars their standard error. The grey points and lines represent each participant's dynamics.

proportion of online gaze visit on session 10 compared to session 1 [.49, $\chi^2(1, N = 142) = 3.44$, $p = .064$] (Fig 8F).

*Gaze duration*. Fig 9 displays the duration of the last gaze visit on handholds before touching them. The Shapiro-Wilk test showed that the assumption of normality was violated for the IVG on session 10. The results of the mixed ANOVA showed a medium effect of the factor Session [$F(1,17) = 10.48$, $p = .005$, $\eta_p^2 = .14$]. The factor Group [$F(2,17) = 1.49$, $p = .253$, $\eta_p^2 = .11$] and the interaction between Session and Group [$F(2,17) = 0.23$, $p = .801$, $\eta_p^2 = .01$] were not significant. The Bayesian mixed ANOVA suggested anecdotal evidence in favor of the null hypothesis for the factor Group ($BF = 0.57$) and the interaction between Session and Group ($BF = 0.46$). The post-hoc test revealed that the duration of the last gaze visit was shorter in session 10 compared to session 1 ($M = -100$ ms, $CI = [-165$ ms, $-35$ ms], $p = .005$).

## Discussion

The aims of this study were to examine how the gaze control of action is adapted to different practice conditions and how changes in gaze control may contribute to the transfer of learning. We measured the climbing fluency and gaze behaviors of novice climbers on a training and transfer route and we compared three practice conditions: constant practice (CG), imposed variable practice (IVG) and self-controlled variable practice (SVG). Results did not show any beneficial effects of IVG on the learners' climbing fluency on the training and transfer routes in comparison with CG. Moreover, CG performed better than IVG on the training route in the last session according to the spatial fluency indicator. The complexity of the gaze path evolved similarly for the three groups on the training and transfer routes with a decrease in variability with practice. Finally, the three groups demonstrated different adaptations of the dual-demand of gaze when controlling their hand movements on the training route at the end of the learning protocol: CG used more online gaze control of their hand movements whereas, in contrast, IVG used more proactive gaze. SVG did not change their gaze pattern, that is they maintained a proportion of .41 of online gaze visits. In addition, on the transfer route, only IVG adapted the gaze control of hand movements in a similar fashion as the last session on the training route.

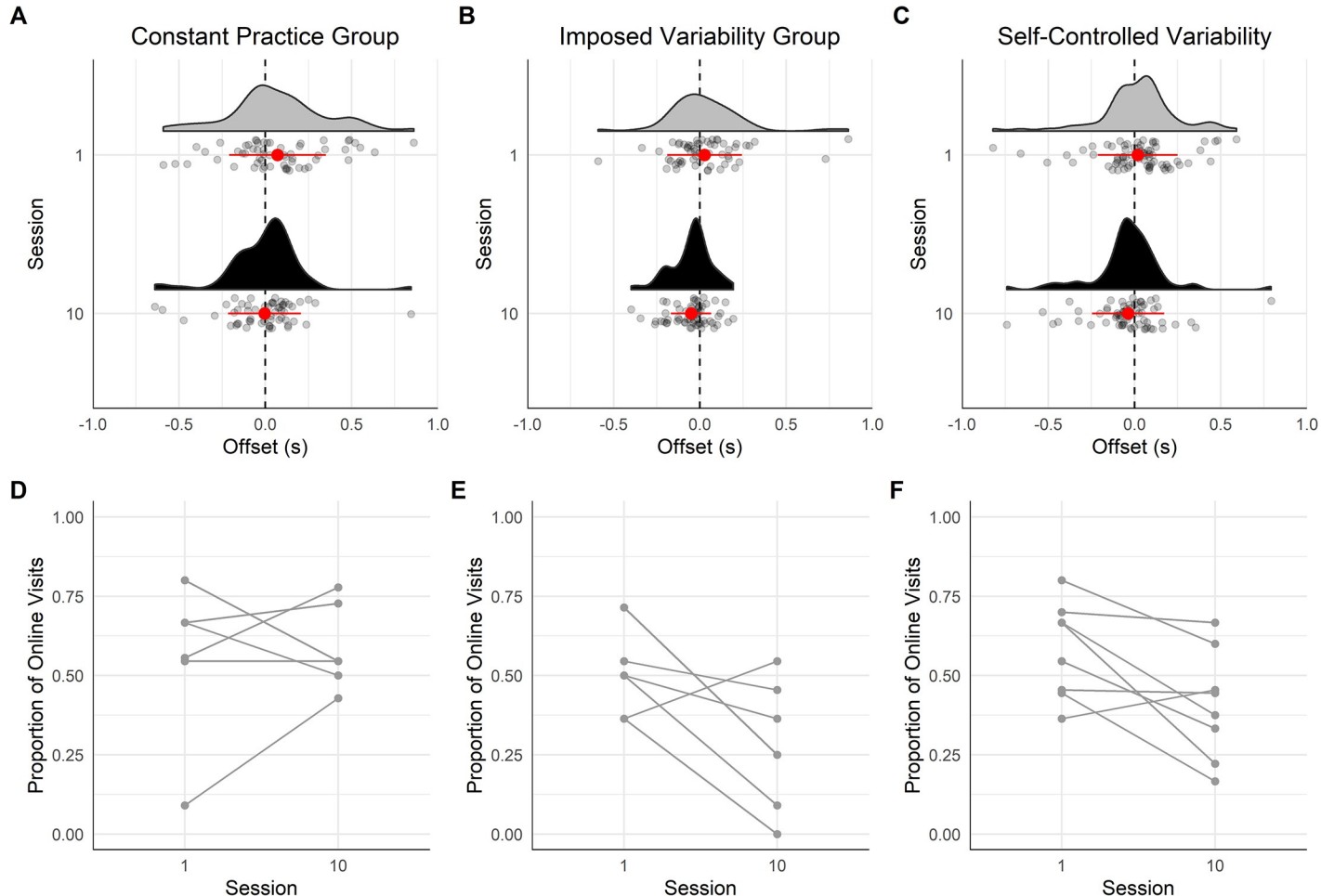

**Fig 8. Offset time of the last gaze visit before the hand contacted the handhold for the three groups on the transfer route.** In (A), (B) and (C), the vertical dashed line shows the time the hand touched the handhold, each point represents one gaze visit, the half violin shows the density of points, the red/grey point with the error bar refers to the mean of all the gaze visits and the standard deviation around the mean. The color of the half violin refers to the learning session: in grey, session 1 and in black, session 10. (D), (E) and (F) displays the individuals' proportion of online visits on session 1 and 10. (A) and (D) show data for the constant practice group, (B) and (E) for the imposed variability group, and (C) and (F) for the self-controlled variability group.

## Climbing fluency

**Specificity of the individual-environment coupling.** In contrast to our hypothesis, results indicated that variable practice conditions did not facilitate transfer to the new climbing route. Previous research in perceptual-motor learning has revealed benefits from variable practice for the transfer of learning [21, 22]. However, the results revealed that all three groups improved their climbing fluency on the transfer route. Therefore, even during CG practice, the findings in the current study suggest that this condition was sufficiently complex to foster adequate exploration. Indeed, the CG practiced on a climbing route that offered a range of opportunities for action, which, in contrast with the virtual learning environments used in the previous studies [21, 22], may have invited more perceptual-motor exploration during practice. Thus, CG may have benefitted from their intrinsic variability to discover a range of information-movement couplings, which facilitated performance on the transfer route.

The benefit of exploration within a constant learning environment is also supported by the lower complexity of the hip trajectory on the last session of CG compared to IVG on the

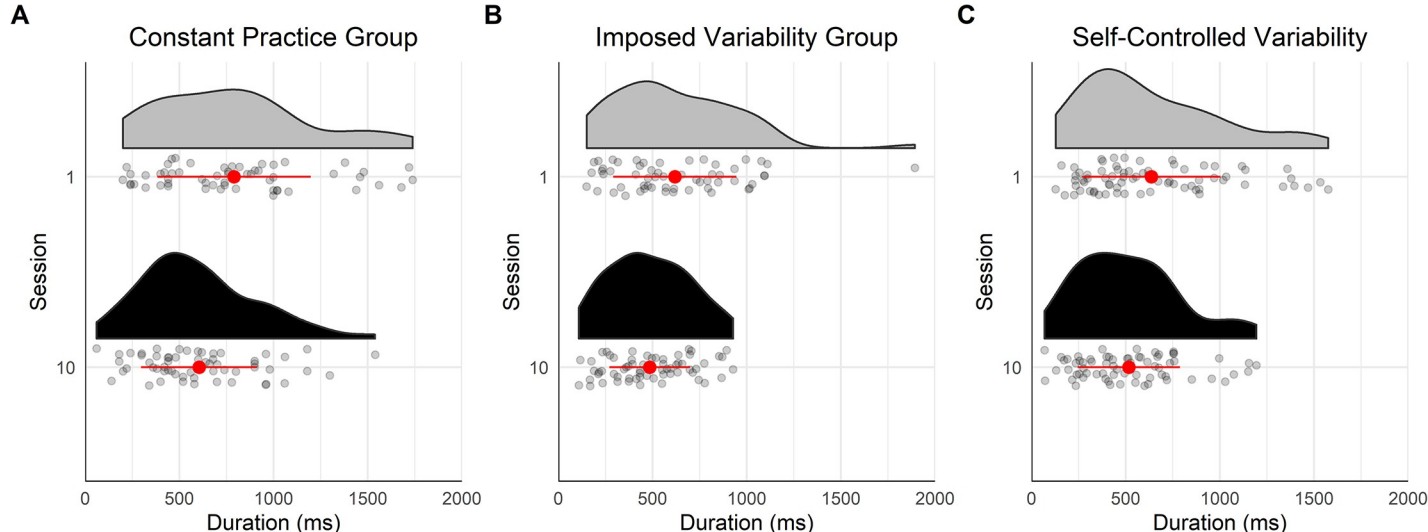

**Fig 9. Duration of the last gaze visit before the hand contact the handhold for the three groups on the transfer route.** In (A), (B) and (C), each point represents one gaze visit before a contact with a handhold, the halves violin shows the density of points, the red/grey point with the error bar refers to the mean of all the gaze visits and the standard deviation around the mean. The color of the half violin refers to the learning session: in grey, session 1 and in black, session 10. (A), (B) and (C) shows data for the constant practice group, imposed variability group and the self-controlled variability group respectively.

training route. This measure was designed to assess the degree of coherence in information-movement couplings [37] and was recently reported to reflect climbing efficiency (i.e., a lower complexity is linked to lower energy expenditure) [48]. Thus, the difference between the two groups suggests that CG benefited from practice on the training route to discover improved information-movement couplings. Indeed, participants in IVG and SVG did not appear to improve their climbing fluency on the training route to the same extent as the CG. Thus, the results suggest that for novice climbers, practice on the variable routes did not adequately transfer to the original training route. Thus, it is plausible that variable practice environments–in climbing and other domains—may be more beneficial for participants with a higher initial skill level.

**No benefit of self-controlled practice on climbing fluency.** Overall, SVG had a lower variability of practice conditions than IVG during their practice, partly because they could only slow the rate of change in practice conditions but not increase it (Table 1). We expected that SVG would better reflect individual learning dynamics, thus, resulting in better learning outcomes than the imposed exploration interventions (IVG and CG). However, no differences in climbing fluency were observed between the three groups at the end of the practice period on the training route and on the transfer route. Moreover, participants chose to keep the same variants on the route in the final sessions as well as the first sessions (Table 2). This observation differs from the practice schedules observed by [27]. In their study, participants controlled their practice between three variations of a key-pressing task, and most participants chose to start with a blocked pattern of practice before finishing with rapid switches between the variations. In the current study, the chosen practice schedules of the SVG suggest that participants were initially more attracted by novelty before practicing a smaller range of climbing routes in the final sessions. This practice structure, from initial variability, reducing to blocked conditions may not be an optimal path for safely exploring and discovering the possible task solutions across the different routes. Therefore, it would be necessary for further research to investigate the effect of early variations on the effectivity of exploration in comparison to variations occurring later in the practice period [49].

The absence of differences in climbing fluency between IVG and SVG contrasts previous research on autonomy-supportive interventions (e.g., [50, 51]) that showed: (i) an increase in performance, even when the choice affects an irrelevant feature of the task (e.g., the color of a golf ball in golf putting, [50]); and (ii) improved performance on a transfer task when participants were given control over their practice schedules [27]. The results of the present study may be due to the duration of the practice period (5 weeks), which is longer and may therefore offer greater insight into the learning process than previous self-controlled research. A review of self-controlled practice interventions [52], revealed that practice took place over four days at maximum and most of the reviewed studies had practice completed in one day. Thus, the five weeks practice schedule in the current study may have diluted the perception of autonomy and associated motivational effect on performances. Moreover, in previous research, the autonomy offered to participants tended to be right after each trial (e.g., they chose to receive feedback or they chose the task condition for the following trial), whereas in the current study, the choice made by the SVG was with reference to the following learning session. Thus, the design of our study may have decreased the participants' perception of control over their learning environment as the delay between the choice and its effect is much larger than in previous studies [52]. This may have prevented the motivational effect of the intervention in the SVG as according to the Control Effect Motivation hypothesis, motivation is sensitive to one' s control over the upcoming events [53, 54]. This potential temporal effect of autonomy on motivation and performance needs to be further investigated.

## Gaze behaviors

**Constant practice conditions foster online gaze control.** The CG relied more on online gaze control of their hand movements than the IVG, who conversely, showed more proactive gaze control with practice on the training route. Previous studies demonstrated that online gaze control supports more accurate stepping behavior [9] and that when walking over rough terrain, participants look a shorter distance ahead than when the terrain is flat [5]. Thus, comparable to these examples, participants in CG tuned their gaze behavior to perform their climbing movements as accurately as possible by exploiting proximal locations of the training route.

Conversely, IVG used more proactive gaze control following practice on the training route. This result showed that IVG participants tuned their gaze behavior to anticipate the constraints on future movements. This kind of gaze pattern has also been observed in natural tasks, when participants perform a series of action during goal achievement [4, 55]. That is, participants look at an object that they are about to manipulate (e.g., a cup) before reaching for it [56]. With such gaze behavior, Land [56] proposed that performers free their visual system as soon as other perceptual systems (e.g., haptic) are in sufficient contact to regulate the movement. Similarly, studies on the development of locomotion showed that infants relied more on online control than children and adults when walking in a room with obstacles [12, 13]. Thus, it appears that the participants in the IVG showed changes in gaze behaviors similar to those observed in the development of locomotion with a gaze behavior that, with practice, is used to guide the hand movements rather than to control them as participants learn to utilize haptic information with practice [57]. These gaze behaviors may support fluent climbing (i.e., chaining climbing actions smoothly); however, it appears that to reach higher climbing fluency, visual information during the contact phase is also necessary.

In the development of locomotion, research has indicated that the gaze behavior of infants was different in quadrupedal locomotion (crawling) than when walking [12]. Moreover, gaze control during quadrupedal locomotion used more online gaze control than walking, in a

similar fashion as reaching for object [12]. Thus, as our study focused on the gaze control of hand movements, our results may not be directly applicable to the control of foot movements while climbing, as the literature suggests that more proactive (or even peripheral) control of action is preferred for foot placements when the task demand allows lesser accuracy demands of foot placement [9, 13]. Future studies may investigate further gaze control of action in climbing to better highlight the differences of how hand and foot movements are controlled with the visual and/or the proprioceptive system (e.g., [58]).

**Proactive gaze control cooperates with the transfer route.** In line with our hypothesis that the CG would develop a gaze pattern highly specific to the training route while IVG would lead to a more adaptive gaze pattern, results showed that the more proactive gaze strategy of the IVG on the training route was also observed on the transfer route. In contrast, the CG showed a similar fixation offset time on session 1 and 10 of the transfer route. Thus, although the climbing fluency measure did not indicate that variable practice facilitated transfer, the gaze behavior developed during practice led to the development of an exploratory (proactive) gaze behavior that was adapted to the task demand of a new climbing route [59, 60]. Conversely, the gaze behavior developed by CG on the training route was not utilized on the transfer route as participants reverted to the gaze behavior used on session 1 of the transfer route. Thus, for CG, the gaze behavior developed with practice competed with the task demand of the new route, compromising general transfer [59, 60]. Considered in tandem with the climbing fluency results, repetition of practice conditions is necessary to improve performance but extensive practice within the same conditions appears to "specialize" gaze behaviors, which limits adaptability. Specialization in this context refers to the learner becoming attuned to information that is variant across other environmental conditions [22]. In contrast, participants in the IVG "learned to explore", that is, they developed a gaze pattern that facilitated the pick-up of information to act adaptively under new constraints [49]. Thus, the present study illustrates that learning and becoming skilled is not only revealed by the performatory activity (i.e., the coordination pattern used and the associated performance) but also by the changes in exploratory behaviors underpinning performatory activity [49].

**The self-controlled group stayed in a comfort zone during practice? Or did they show different individual benefits?.** Although practicing on the variable routes, the participants in SVG did not significantly change the time of fixation offset. Therefore, in comparison with IVG participants who adapted their gaze behavior, SVG did not adjust gaze patterns, which may be a function of the comparative decrease in the amount of variability during practice for SVG. This result suggests that participants in SVG chose (i.e., whether to train on a new route or not) to remain within a comfort zone during practice [25]. However, when individual data is considered, the effects of practice on the training route are distinct between SVG participants (Fig 4) with five of the eight participants clearly showed less online gaze control of their climbing movements on the transfer route (Fig 8). Thus, rather than all participants remaining in a comfort zone, it appears that on the training route, some developed a gaze behavior similar to CG while others were similar to IVG. These interindividual differences highlight the importance of considering variability in gaze patterns [61], revealing that some participants may have developed both gaze patterns and used them adaptively according to the task demands. Although, we did not manipulate the route difficulty, the confrontation to a new route is in itself more difficult than climbing a known route. Self-controlled practice may therefore represent a means to enable individualization of how challenging the learning environment is during practice [25, 26]. Further investigation on the individual learning pathways may be productive in better understanding how learners can benefit most from such interventions.

## Summary and future directions

This study followed previous work that acknowledged the importance of the timing of the information pick-up in complex perceptual-motor tasks [62]. The findings of the present study highlighted that information pick-up was affected by practice conditions. This may explain why variable practice facilitates transfer of learning while constant practice may lead to a specialization of individuals to the learning task constraints. However, the location of the point of gaze with eye-tracking technology does not necessarily capture what information is used for movement control, whilst the practice schedules may have differentially affected participants' attunement. Extant literature indicates that behavior and performances in complex perceptual-motor tasks follows a nonlinear learning process [63–65]. In contrast, the learning dynamics of changes in gaze patterns during learning remains relatively unexplored [49]. Thus, the results from the current study, particularly given the interindividual variability in SVG suggests that a challenge for future investigations is to reveal the individual learning dynamics of information pick-up in tandem with behavioral dynamics. We may expect to observe different gaze patterns to achieve similar task outcomes within and between individuals, highlighting degeneracy in perceptual-motor control [61]. Such investigation may further understanding of why some individuals develop better adaptability than others, even when exposed to the same intervention.

## Supporting information

**S1 Data. Tabular presentation of the dependent variables.**
(ZIP)

## Acknowledgments

We thank Héloïse Baillet for her contribution to the data collection and treatment. We also thank the participants for their cooperation and involvement in the study.

## Author Contributions

**Conceptualization:** Guillaume Hacques, Matt Dicks, John Komar, Ludovic Seifert.

**Formal analysis:** Guillaume Hacques, John Komar.

**Funding acquisition:** Ludovic Seifert.

**Investigation:** Guillaume Hacques.

**Methodology:** Guillaume Hacques, Matt Dicks, Ludovic Seifert.

**Project administration:** Ludovic Seifert.

**Supervision:** John Komar, Ludovic Seifert.

**Visualization:** Guillaume Hacques.

**Writing – original draft:** Guillaume Hacques, Matt Dicks.

**Writing – review & editing:** John Komar, Ludovic Seifert.

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
