## [Decision Letter · Decision Letter 0]

2 Mar 2022

PONE-D-22-01772Visual control during climbing: Variability in practice fosters a proactive gaze patternPLOS ONE

Dear Dr. Hacques,

Thank you for submitting your manuscript to PLOS ONE. After careful consideration, we feel that it has merit but does not fully meet PLOS ONE’s publication criteria as it currently stands. Therefore, we invite you to submit a revised version of the manuscript that addresses the points raised during the review process. Most importantly, one reviewer highlights a potential issue with the sample size calculation and suggests the possibility that the study is underpowered. 

We look forward to receiving your revised manuscript.

Kind regards,

Greg Wood, PhD

Academic Editor

PLOS ONE

Journal Requirements:

Reviewers' comments:

Reviewer's Responses to Questions

**Comments to the Author**

1. Is the manuscript technically sound, and do the data support the conclusions?

Reviewer #1: Yes

Reviewer #2: Yes

2. Has the statistical analysis been performed appropriately and rigorously? 

Reviewer #1: Yes

Reviewer #2: No

3. Have the authors made all data underlying the findings in their manuscript fully available?

Reviewer #1: Yes

Reviewer #2: Yes

4. Is the manuscript presented in an intelligible fashion and written in standard English?

Reviewer #1: Yes

Reviewer #2: Yes

5. Review Comments to the Author

Reviewer #1: The manuscript describes a technically sound piece of scientific research that addresses a previously under-researched area of study. The study is novel, in that it extends current body of research related to practice schedules and the acquisition of climbing skill. Furthermore, the study adopts a research paradigm that has previously been under-utilised within the study of climbing performance, combining mobile eye-tracking technology with pressure-sensitive climbing holds and GIE data. Previous research utilising mobile eye-tracking technology has been criticised for being reliant upon aggregated gaze behaviours and not adequately capturing the temporal aspect of participant’s gaze behaviour, due to the labour-intensive nature of manual analysis. One of the distinct strengths of this study, is combining multiple complimentary measures: Aligning gaze data with the timestamps from pressure-sensitive holds, enables the temporal aspect of gaze behaviour in climbing performance is successfully captured. In this respect, gaze behaviour is captured in the context in which it occurred. The further addition of GIE data enables inferences to be made as to not only which form of practice schedule is most effective (increased fluency), but also make inferences about which gaze strategies might be more effective.

A succinct rationale for the study is provided, drawing upon relevant literature to draw parallels between the attentional demands of climbing movement, in regards to the visual search strategies adopted, and the attentional demand of walking/climbing. Whilst this comparison is valid in many respects, the manuscript may have benefitted further from a fuller discussion of the performance demands of climbing - how they differ from walking/running (e.g. multiple points of contact used in conjunction, more 3D in how forces are applied, etc) and how this inevitably impacts attentional focus and gaze behaviour.

The cognitive processes involved whilst engaged in climbing performances are perhaps oversimplified, categorising attentional focus as a dichotomy between maintaining current movement and monitoring the environment for information that may constrain future movements. This may potentially disregard a wide range of other factors (e.g. thoughts of anxiety, apprehension) that may impact upon attentional focus and gaze behaviour. Furthermore, the embodied nature of gaze behaviour might also be worth further exploration. For example, as climbing movements become practised does visual search become more intuitive, and not consciously deliberative? There is a brief mention of this in lines 713-716, but could be elaborated further. Finally, whilst I am personally an advocate for the utility of eye-tracking technology, recording the point of gaze does not necessarily capture what information is drawn from that location and how that relates to cognition. A brief reference to these limitations might, therefore, be appropriate.

The report adheres to standard reporting guidelines, providing a good level of detail on the study’s proposed methodology for data collection and analysis. The methods section provides a generally excellent level of detail giving clear consideration to considerations of reliability and future replication. There were a few small areas that would benefit from additional detail or clarification. Firstly, one of the main challenges with using mobile eye-tracking technology in real-world climbing tasks is the limited trackable visual range (vertical 52 deg) of the glasses, as compared to the normal human field of vision (vertical 125 deg). Given that climbers are in close proximity to the wall, they are presumably required to use the extreme vertical range of their visual field in order to see prospective holds above and their feet placement below, perhaps exceeding the trackable range of the glasses. Would the researchers clarify whether there was any additional instruction provided to the participants to circumnavigate this limitation? If so, it might be worth detailing in the procedures section.

The methods for analysing the GIE, gaze, and Luxov Touch data were all appropriate, described to a high level of detail, and conducted to a high technical standard. One small area for clarification might be how gaze behaviour for ‘online movements’ were analysed. The defined areas of interest were clear, but I think you stated that you only focused on hand holds, presumably omitting some fixations that were focused on feet movements or prospecting foot holds? The rationale for adopting this approach were not 100% clear, nor how it may have impacted the proportion of time split between online/proactive gaze control.

The practice schedules for the three control groups were clearly outlined and logically structured. Perhaps the only small detail omitted was the rest period between attempts, as this has implications for time available for route reading from the ground. One small area for concern was in the instruction to participants to ‘not pause’ when climbing, as this strategy is not representative of what would be considered ‘normal’ climbing. Most climbers will commonly take any resting opportunity to not only recover, but also to route read ahead to plan their next series of movements. In this respect, the experiment may encourage a gaze strategy that is not representative of normal climbing performance. This said, the manuscript provides more than sufficient detail for the reader to decide the validity and transferability of the results.

Reviewer #2: The authors studied whether different practice schedules would differentially affect gaze behavior in climbing. This manuscript represents a huge effort to address their goal: many hours of data collection, long-term practice, and appropriate design (pretest transfer, posttest transfer, etc). I applause the authors on this.

Major:

- My main concern reflects the issue of sample size. Despite the authors stating that they calculated the required sample size through GPower, I believe that there is a mistake in their calculation. As far as I am aware (also see Faul et al., 2007, Behavior Research Methods, 39(2), 175-191), the value that one must include in "number of measurements" is the number of repeated measures and, in this case, it should be 2 (not multiplying it by number of groups, as the authors performed). Doing the calculation for 80% power, medium effect size, 3 groups, and 2 measurements leads to 42 participants (not 24). The actual power is 52% (if I am correct). In the end, having only 21 participants, this decreases even further to 46%.

The only power that they are guaranteed to find (with 80%) is a large effect size.

I understand that having to collect more data is always an issue (lines 420-427), especially when the data collection is so demanding. However, I would not use this as an argument to not achieve the required sample size for the effects I would like to observe (even more when considering that the paper was funded).

- Another concern that is spread over the paper is that the authors have the tendency to bring strong statements that are not fully supported by the literature. This is problematic and, in my opinion, not even necessary for the paper. For instance, the authors claim that instrinsic variability is insufficient for learners to progress in learning which I (see below) cannot agree. Other instance, in the discussion, the authors claim that previous research in perceptual-motor learning has revealed benefits from variable practice for transfer. This, again, is not true in general. There are examples through all introduction and discussion (even though I mostly highlighted the ones in the intro). My suggestion would be to revise the paper and add the nuances here and there that the literature demonstrates.

Other than that, the paper has only minor issues:

- Line 59 - "The performer must find a trade-off" . I would change trade-off for "balance".

- Line 78 - No need for "however" here.

- Lines 74 and 84 - Tt seems that refs (9) and (12,13) are opposing findings regard the same age range (young adults). If I understand correctly, the task dimension is important to determine when one type of gaze is important or not. This should be more explicitly stated.

- Line 110 - The (16) paper did show that transfer depends on exploration - systematic change over time - rather than variability. In fact, no measure of variability (in the sense of data variance or spread) was maintained in their prediction of transfer performance. You could say that different practice schedules have the potential to differentially guide exploration.

Lines 114-115: This statement is wrong. If the authors want to mean that constant practice is insufficient, there are many examples of studies with tasks that have a single condition that leads to change in initial behavioral tendencies. In fact, the main theoretical formalisms of dynamical systems were considered in terms of a system practicing a single task condition (see Schoner et al., 1992; Zanone et al., 1992). If the authors want to mean that intrinsic variability is insufficient for changing initial tendencies, it is also problematic. Initial tendencies in tasks that require new movement patterns lead to great competition between organism and task. This, in turn, leads to increased variability (arising from the system). This, as stated above, would be sufficient to lead the system to a new solution and, thus, "progress from their initial behavioral tendencies". My suggestion would be to direct the argument in terms of "differential" guidance of exploration rather than "insufficient variability". In fact, this would be in line to (17) who proposed different goal spaces for variable and constant practice.

- Lines 153-156 - As far as I am aware, self-controlled studies have shown, mostly, results in transfer tests. Not much during practice (or retention). From your rationale, the performance during acquisition should also differ. How can your explanation deal with the rest of the literature?

- Lines 172-173: So far, the argument was that, if the variations allow perception of more useful information, then transfer would be better (if the given information was also useful in this new context). Now, it became that variable practice leads to more adaptable behaviors - which is wrong (see van Rossum, 1990; or even Pacheco et al., 2018 - Experimental Brain Research). I would delete this sentence. The paragraph does not need that statement.

- Lines 338-340: I am not sure that more complex hip trajectory leads to "poor sensitivity of the climber to the environmental constraints". more or less complexity depends on the task (as highlighted by Newell & Vaillancourt, 2001 - Human Movement Science). I would add a rationale supporting this expectation in this task.

- Statistical Analyses: I do not understand why the authors do not use JUST the Bayesian statistics. The idea, usually, is that the Bayesian approach represents another approach to statistical analyses (not a complementary one). Using both does not make much sense as they have different "rationales" or "physolophical assumptions".

- Line 447 - "Error! Reference source not found"

- Lines 633-636: I do not think the authors provided sufficient rationale to point out that the variable practice provided in the present paper would be sufficient to attune individuals to A higher-order information that would facilitate transfer. The issue that I am raising comes from the fact that not all variable practice should be related to all transfer tests and not all variable practice should lead to attunement - considering the direct learning idea or any other approach.

- Line 639: It could be transfer was not sufficiently complex to need exploration in practice. How exploration was measured?

- Lines 642-643: In the intro, the authors claimed that intrinsic variability was not enough. Now, it is.

- Lines 659-661: I believe it is one thing to claim that there are intrinsic dynamics that affect what is the best for a given individual in a given task and another to claim that individuals would be aware what is best for them. The authors assumed both - why would that be the case?

- Lines 741-743 - Did the CG showed overspecialization? Or any decrement to support the statement of lack of "general transfer"?

I hope my comments help the authors.

6. PLOS authors have the option to publish the peer review history of their article (what does this mean?). If published, this will include your full peer review and any attached files.

Reviewer #1: **Yes: **James Mitchell

Reviewer #2: No

---

## [Author Response · Author response to Decision Letter 0]

15 Apr 2022

Dear Dr. Wood, Dear Reviewers,

We would like to thank the reviewers for their helpful comments, which provided us the opportunity to improve the manuscript. We addressed all the reviewers’ comments and questions. You can find in the text below our point-by-point responses (marked with *). The indicated line numbers correspond to those of the manuscript with marked changes (changes are highlighted in red).

We hope that our responses and rewrite will meet your expectations.

With best regards

Reviewer #1: 

- The manuscript describes a technically sound piece of scientific research that addresses a previously under-researched area of study. The study is novel, in that it extends current body of research related to practice schedules and the acquisition of climbing skill. Furthermore, the study adopts a research paradigm that has previously been under-utilised within the study of climbing performance, combining mobile eye-tracking technology with pressure-sensitive climbing holds and GIE data. Previous research utilising mobile eye-tracking technology has been criticised for being reliant upon aggregated gaze behaviours and not adequately capturing the temporal aspect of participant’s gaze behaviour, due to the labour-intensive nature of manual analysis. One of the distinct strengths of this study, is combining multiple complimentary measures: Aligning gaze data with the timestamps from pressure-sensitive holds, enables the temporal aspect of gaze behaviour in climbing performance is successfully captured. In this respect, gaze behaviour is captured in the context in which it occurred. The further addition of GIE data enables inferences to be made as to not only which form of practice schedule is most effective (increased fluency), but also make inferences about which gaze strategies might be more effective.

- A succinct rationale for the study is provided, drawing upon relevant literature to draw parallels between the attentional demands of climbing movement, in regards to the visual search strategies adopted, and the attentional demand of walking/climbing. Whilst this comparison is valid in many respects, the manuscript may have benefitted further from a fuller discussion of the performance demands of climbing - how they differ from walking/running (e.g. multiple points of contact used in conjunction, more 3D in how forces are applied, etc) and how this inevitably impacts attentional focus and gaze behaviour.

*We added l.91 to 96 a description of the performance demands in climbing: and how it impacts gaze behaviour : “Climbing locomotion is performed on a vertical plane and climbers need to find support surfaces on the wall (the handholds and footholds) on which to apply forces to climb up the route while maintaining their balance with one to four points of contact (the hands and feet) (14). In this context, the visual system is used (i) to locate the support surfaces on the wall, and (ii) to perceive and act on the opportunities for action that these supports and their configuration on the wall afford to the climber.”

- The cognitive processes involved whilst engaged in climbing performances are perhaps oversimplified, categorising attentional focus as a dichotomy between maintaining current movement and monitoring the environment for information that may constrain future movements. This may potentially disregard a wide range of other factors (e.g. thoughts of anxiety, apprehension) that may impact upon attentional focus and gaze behaviour. 

*We go beyond the presented dichotomy in the results as we consider some other aspects of gaze behavior (although they were not central to the paper research question) by measuring the visual entropy and duration of the gaze visit of the handholds, as gaze behavior can also be used to look for the chain of actions and to investigate the holds respectively. We also agree with the reviewer’s comment that gaze behavior in climbing task can be affected by other cognitive factors. For example, anxiety was shown to affect gaze behavior in climbing (Nieuwenhuys, Pijpers, Oudejans & Bakker, 2008). As we were aware of this potential effect, we designed the climbing task to limit this effect as much as possible: the maximum height of the routes was about 5m (which is low for a climbing route) and the participants were climbing top-roped (which is less engaging than lead climbing). 

- Furthermore, the embodied nature of gaze behaviour might also be worth further exploration. For example, as climbing movements become practised does visual search become more intuitive, and not consciously deliberative? There is a brief mention of this in lines 713-716, but could be elaborated further. 

*This is an interesting question that we cannot respond based on our measures. The change in gaze behavior that we observed in this study should be implicit to the learning protocol that we designed. Indeed, the participants were not instructed to perform any of the observed gaze behaviors: the observed changes should be due to the different practice schedules. We would expect that their attentional focus would be directed toward the task goal (i.e., climbing as fluently as possible avoiding pauses and jerky movements). Thus, we would expect that the participants would never “consciously” perform one form of gaze behavior or another, but that the participants implicitly “organize” their gaze behavior with practice to improve the chaining of their actions on the climbing routes.

- Finally, whilst I am personally an advocate for the utility of eye-tracking technology, recording the point of gaze does not necessarily capture what information is drawn from that location and how that relates to cognition. A brief reference to these limitations might, therefore, be appropriate.

*We added a sentence about these limitation in the “Summary and future directions” section l.782-785 : “However, the capture of the location of the point of gaze with eye-tracking technology does not necessarily capture what information is used for movement control, whilst the practice schedules may have differentially affected participants’ attunement.”

- The report adheres to standard reporting guidelines, providing a good level of detail on the study’s proposed methodology for data collection and analysis. The methods section provides a generally excellent level of detail giving clear consideration to considerations of reliability and future replication. There were a few small areas that would benefit from additional detail or clarification. Firstly, one of the main challenges with using mobile eye-tracking technology in real-world climbing tasks is the limited trackable visual range (vertical 52 deg) of the glasses, as compared to the normal human field of vision (vertical 125 deg). Given that climbers are in close proximity to the wall, they are presumably required to use the extreme vertical range of their visual field in order to see prospective holds above and their feet placement below, perhaps exceeding the trackable range of the glasses. Would the researchers clarify whether there was any additional instruction provided to the participants to circumnavigate this limitation? If so, it might be worth detailing in the procedures section.

*Indeed, the limited trackable visual range is one of the limitations in the use of eye-tracking system in sporting tasks. In climbing, and specifically with the Tobii 2 eye tracking glasses, the point of gaze was lost when the participants were looking toward their feet while standing. As capturing the gaze related to the feet movements would have required to constrain the visual field of the climber, or to instruct them to perform unnatural head movements, we chose to focus on the gaze behavior related to hand movements. However, on session 10 we observed that some participants of the constant practice group were never looking down at their feet during their climb of the control route, which suggested that, whilst they used more online control of their hand movements, they did not use foveal visual information to control their feet movements. Thus, as we stated l.725 to 734, whilst investigating the differences in the visual control of hand and feet movements in quadrupedal locomotion such as climbing is definitely an interesting line of study, given the current state of eye tracking technology, it would be necessary to apply additional constraints to the visual system of the participants.

- The methods for analysing the GIE, gaze, and Luxov Touch data were all appropriate, described to a high level of detail, and conducted to a high technical standard. One small area for clarification might be how gaze behaviour for ‘online movements’ were analysed. The defined areas of interest were clear, but I think you stated that you only focused on hand holds, presumably omitting some fixations that were focused on feet movements or prospecting foot holds? The rationale for adopting this approach were not 100% clear, nor how it may have impacted the proportion of time split between online/proactive gaze control.

*We focused on the gaze behavior related to hand movements as explained in the previous response to the reviewer’s comment. More precisely, “online” and “proactive” gaze were measure based on the gaze offset time, that is the time difference between when the handhold was touched and when the point of gaze moved away from the handhold (l. 362-364). 

- The practice schedules for the three control groups were clearly outlined and logically structured. Perhaps the only small detail omitted was the rest period between attempts, as this has implications for time available for route reading from the ground. One small area for concern was in the instruction to participants to ‘not pause’ when climbing, as this strategy is not representative of what would be considered ‘normal’ climbing. Most climbers will commonly take any resting opportunity to not only recover, but also to route read ahead to plan their next series of movements. In this respect, the experiment may encourage a gaze strategy that is not representative of normal climbing performance. This said, the manuscript provides more than sufficient detail for the reader to decide the validity and transferability of the results.

*As indicated l.285, the participants had a maximum of 30s to read the route for all trials: “(iii) the participant stood 3m in front of the route for 30s of route preview.” The routes that were not climbed were hidden with a tarpaulin to prevent additional route previewing. Thus, regarding the rest period, the participant had the 30s of previewing the route and approximately 1min while the experimenter calibrated the eye-tracker, repeated the instructions, uncovered the route and checked the safety rope. These steps are described l.283 to 295.

*Regarding the climbing task used for the experiment, these instructions are commonly used in studies about perceptual-motor learning in climbing. The instruction “to chain movements without pauses” was designed to challenge the participants throughout the learning protocol and to observe to what extent they could perceive and act on nested affordances with practice (for a review on the topic, see Orth, Davids & Seifert, 2015, Sport Medicine). 

Reviewer #2: The authors studied whether different practice schedules would differentially affect gaze behavior in climbing. This manuscript represents a huge effort to address their goal: many hours of data collection, long-term practice, and appropriate design (pretest transfer, posttest transfer, etc). I applause the authors on this.

Major:

- My main concern reflects the issue of sample size. Despite the authors stating that they calculated the required sample size through GPower, I believe that there is a mistake in their calculation. As far as I am aware (also see Faul et al., 2007, Behavior Research Methods, 39(2), 175-191), the value that one must include in "number of measurements" is the number of repeated measures and, in this case, it should be 2 (not multiplying it by number of groups, as the authors performed). Doing the calculation for 80% power, medium effect size, 3 groups, and 2 measurements leads to 42 participants (not 24). The actual power is 52% (if I am correct). In the end, having only 21 participants, this decreases even further to 46%.

The only power that they are guaranteed to find (with 80%) is a large effect size.

I understand that having to collect more data is always an issue (lines 420-427), especially when the data collection is so demanding. However, I would not use this as an argument to not achieve the required sample size for the effects I would like to observe (even more when considering that the paper was funded).

*Thank you for highlighting this point as it appears that we made a mistake in our calculation on GPower. As observed by the reviewer, the number of measurements should be 2. In an earlier version of this manuscript, we used a linear mixed model to analyze the effect of Groups, Session and Trial on our main dependent variable (that is the Gaze offset time), which enabled us to avoid averaging this data to perform a mixed-ANOVA, which is presented in the current version of the manuscript. Moreover, our “raw” dataset – e.g., the “offset_duration.csv” file in the S1_Data folder of the Supporting Information– on the control route includes approximately 33 measurements per session and per participant. However, after discussing our analysis with a researcher external to the project, it was recommended that we simplify the statistical model used and to add the Bayesian analysis to give more information relative to the non-significant results. As such, the linear mixed model previously used showed that same results as the current mixed-ANOVA (that is only a significant Group x Session interaction):

*“The final linear mixed model included participant and handhold as adjustments of the intercept. The fit of the LMM applied to time of the gaze visit offset, was improved by the interaction between Session and Group [LLR χ² (2) = 51.41, p < .001]. The fixed effects Group [LLR χ² (2) = 5.49, p = .064], Trial [LLR χ² (2) = 0.50, p = .777], Session [LLR χ² (1) = 0.62, p = .430], the interaction between Session and Trial [LLR χ² (2) = 2.00, p = .368], Trial and Group [LLR χ² (4) = 2.27, p = .686] and between Group, Session and Trial [LLR χ² (4) = 0.404, p = .982] did not significantly affect the LMM fit.”

*Thus, the study design that we initially planned had 6 within participants measurements (3 Trials x 2 Sessions) for our main dependent variable of interest as we indicated in the GPower analysis. Also, the results showed a medium effect size according to the generalized eta squared of the Session x Group interaction on this variable, giving an observed power >.80. These results would still need to be replicated to be confirmed as this study is the first to propose this analysis of the gaze behavior in a climbing task, but now this study can provide a first value of effect size that can be expected.

- Another concern that is spread over the paper is that the authors have the tendency to bring strong statements that are not fully supported by the literature. This is problematic and, in my opinion, not even necessary for the paper. For instance, the authors claim that instrinsic variability is insufficient for learners to progress in learning which I (see below) cannot agree. Other instance, in the discussion, the authors claim that previous research in perceptual-motor learning has revealed benefits from variable practice for transfer. This, again, is not true in general. There are examples through all introduction and discussion (even though I mostly highlighted the ones in the intro). My suggestion would be to revise the paper and add the nuances here and there that the literature demonstrates.

*We changed the sentence relative to intrinsic variability as we agree that it was not appropriate and could be misunderstood: “When the same practice condition is repeated, variability in the performed movement has been found to occur from one repetition to the next. This variability is intrinsic to the motor system.”

*We also made changes to the manuscript in the introduction and discussion according to the reviewer’s comments 

Other than that, the paper has only minor issues:

- Line 59 - "The performer must find a trade-off" . I would change trade-off for "balance".

*We changed as suggested.

- Line 78 - No need for "however" here.

- Lines 74 and 84 - Tt seems that refs (9) and (12,13) are opposing findings regard the same age range (young adults). If I understand correctly, the task dimension is important to determine when one type of gaze is important or not. This should be more explicitly stated.

*The refs (9) and (12,13) are not opposing findings. The observed gaze behavior is different for young adults in the two studies because the task demand is different. In (9), the task was walking while stepping on the center of the target (the task goal was to be as accurate as possible) whereas in (12,13), there was no demand for accuracy. The participants were just walking in a room with obstacles. From the l.64 to 77, we try to focus on conditions where online control is required whereas the following paragraph focus on conditions where proactive gaze behavior is possible.

- Line 110 - The (16) paper did show that transfer depends on exploration - systematic change over time - rather than variability. In fact, no measure of variability (in the sense of data variance or spread) was maintained in their prediction of transfer performance. You could say that different practice schedules have the potential to differentially guide exploration.

*Thank you - we changed as suggested by the reviewer (l.114-116): “According to learning approaches rooted in dynamical system theory, different practice schedules have the potential to differentially guide exploration, which could affect the transfer of learning (16)” 

-Lines 114-115: This statement is wrong. If the authors want to mean that constant practice is insufficient, there are many examples of studies with tasks that have a single condition that leads to change in initial behavioral tendencies. In fact, the main theoretical formalisms of dynamical systems were considered in terms of a system practicing a single task condition (see Schoner et al., 1992; Zanone et al., 1992). If the authors want to mean that intrinsic variability is insufficient for changing initial tendencies, it is also problematic. Initial tendencies in tasks that require new movement patterns lead to great competition between organism and task. This, in turn, leads to increased variability (arising from the system). This, as stated above, would be sufficient to lead the system to a new solution and, thus, "progress from their initial behavioral tendencies". My suggestion would be to direct the argument in terms of "differential" guidance of exploration rather than "insufficient variability". In fact, this would be in line to (17) who proposed different goal spaces for variable and constant practice.

*Indeed, this sentence was not appropriate in this context and could be misunderstood. Thus, we removed the second part stating that intrinsic variability was “insufficient for learners to progress from their initial behavioral tendencies” (l.120-122). We also moved and modified the sentence “This consists in adding unstructured variability to practice at the level of multiple task parameters (18).” (l.126-127) so that now the paragraph gives more clearly the definition of intrinsic and unstructured variability.

- Lines 153-156 - As far as I am aware, self-controlled studies have shown, mostly, results in transfer tests. Not much during practice (or retention). From your rationale, the performance during acquisition should also differ. How can your explanation deal with the rest of the literature?

*From our rationale, the performance should differ both during acquisition and transfer as illustrated by the examples from refs (24) and (26). A review on the topic by Sanli et al. (2013, Frontiers in Psychology) pointed out that self-controlled practice benefitted transfer but also immediate and delayed retention. This review also showed that transfer tests are not the most common test used to assess the effect of self-controlled practice when control of the practice schedule is given. Ref 26 is one of the rare studies using a transfer test. 

- Lines 172-173: So far, the argument was that, if the variations allow perception of more useful information, then transfer would be better (if the given information was also useful in this new context). Now, it became that variable practice leads to more adaptable behaviors - which is wrong (see van Rossum, 1990; or even Pacheco et al., 2018 - Experimental Brain Research). I would delete this sentence. The paragraph does not need that statement.

*This sentence was removed as suggested (l.179-181). 

- Lines 338-340: I am not sure that more complex hip trajectory leads to "poor sensitivity of the climber to the environmental constraints". more or less complexity depends on the task (as highlighted by Newell & Vaillancourt, 2001 - Human Movement Science). I would add a rationale supporting this expectation in this task.

*The Geometric Index of Entropy is very often used in the context of the study of perceptual-motor behaviors in climbing. In the context of this sentence, the adjective “complex” may not be the best suited as we meant a “noisy” or “random” hip trajectory. As the reviewer emphasized, this measure can be task-sensitive: the values should differ from one climbing route to another if the path of the hip is also expected to differ. However, if only the holds are modified between the routes (not their locations), this measure enables researchers to assess the participant sensitivity to the route constraints. In the context of the current study, as the GIE scores are only compared within the same climbing route at different time periods (session 1 and 10), the measure reflects how well participants are able to chain their movements on the route before and after practice.

- Statistical Analyses: I do not understand why the authors do not use JUST the Bayesian statistics. The idea, usually, is that the Bayesian approach represents another approach to statistical analyses (not a complementary one). Using both does not make much sense as they have different "rationales" or "physolophical assumptions".

*We added the Bayesian statistics as suggested by a researcher external to the project. It was recommended that nonsignificant results in inferential statistics should be checked with Bayesian statistics for drawing conclusions. Such statistical approach has already been used in other related studies (e.g., Iodice et al. 2019, PNAS) as proposed by Dienes (2014, Frontiers in Psychology).

- Line 447 - "Error! Reference source not found"

*This was removed.

- Lines 633-636: I do not think the authors provided sufficient rationale to point out that the variable practice provided in the present paper would be sufficient to attune individuals to A higher-order information that would facilitate transfer. The issue that I am raising comes from the fact that not all variable practice should be related to all transfer tests and not all variable practice should lead to attunement - considering the direct learning idea or any other approach.

*We wanted to emphasize here that we varied the layout of handholds of the routes in variable practice conditions so that participants would learn to perceive the chaining of actions from the layout of handholds (for example, from the patterns that the holds shape on the wall). However, it is true that we don’t know what would be the higher order informational variables that would specify how to use the handholds. This is a question that we wanted to address in this project but for which we still need more investigations. Thus, we removed this sentence to avoid speculations about the results of this study (l.639-642). 

- Line 639: It could be transfer was not sufficiently complex to need exploration in practice. How exploration was measured?

*In this sentence, we refer to exploration as changes in movement performance during practice. This sentence is clearly speculating about what could explain that the three groups improved their climbing fluency on the transfer route. The tasks used in motor learning studies are usually much less complex, so we wanted to stress that the nature of the task that we used may already have provided participants with the opportunity to discover various movement solutions (even in a constant practice condition and even if we the designed climbing task was simplified in comparison to a real one). The specific study of exploration on the control route for the three groups is planned in a future research project.

- Lines 642-643: In the intro, the authors claimed that intrinsic variability was not enough. Now, it is.

*We modified the sentence in the introduction as it seemed confusing. We acknowledge that this sentence was not appropriate.

- Lines 659-661: I believe it is one thing to claim that there are intrinsic dynamics that affect what is the best for a given individual in a given task and another to claim that individuals would be aware what is best for them. The authors assumed both - why would that be the case?

*We expected that individuals in the SVG would benefit from the control they were given over their practice schedule because previous studies that we presented in the introduction suggested this would be the case. Notably, Liu et al. (2012) showed that by giving control to the learners over their practice schedule, this appeared to enable them to challenge themselves more optimally than when the practice schedules induced regular changes in task difficulty. Similarly, the results of Wu and Magill (2011) suggested that learners in a self-controlled group had better results in transfer test because they could shape their practice schedule in a more optimal way. 

- Lines 741-743 - Did the CG showed overspecialization? Or any decrement to support the statement of lack of "general transfer"?

*The participants in CG changed their gaze behavior on the control route with practice but on the transfer route, the gaze behavior did not appear to change between session 1 and 10. This result is why we stated that CG “overspecialize” their gaze behavior to the control route and that there was no transfer of the gaze pattern to the transfer route. As “overspecializing” may suggest an associated decrease in performance, we changed the manuscript to say that CG “specialize” their gaze behavior (l.749, l.750 and l.782). 

-I hope my comments help the authors.

*We want to thank the two reviewers for their helpful comments that we think have contributed to improve this manuscript.

---

## [Decision Letter · Decision Letter 1]

31 May 2022

Visual control during climbing: Variability in practice fosters a proactive gaze pattern

PONE-D-22-01772R1

Dear Dr. Hacques,

We’re pleased to inform you that your manuscript has been judged scientifically suitable for publication and will be formally accepted for publication once it meets all outstanding technical requirements.

Kind regards,

Greg Wood, PhD

Academic Editor

PLOS ONE

Additional Editor Comments (optional):

Reviewers' comments:

Reviewer's Responses to Questions

**Comments to the Author**

1. If the authors have adequately addressed your comments raised in a previous round of review and you feel that this manuscript is now acceptable for publication, you may indicate that here to bypass the “Comments to the Author” section, enter your conflict of interest statement in the “Confidential to Editor” section, and submit your "Accept" recommendation.

Reviewer #1: All comments have been addressed

Reviewer #2: All comments have been addressed

2. Is the manuscript technically sound, and do the data support the conclusions?

Reviewer #1: Yes

Reviewer #2: Yes

3. Has the statistical analysis been performed appropriately and rigorously? 

Reviewer #1: Yes

Reviewer #2: I Don't Know

4. Have the authors made all data underlying the findings in their manuscript fully available?

Reviewer #1: Yes

Reviewer #2: Yes

5. Is the manuscript presented in an intelligible fashion and written in standard English?

Reviewer #1: Yes

Reviewer #2: Yes

6. Review Comments to the Author

Reviewer #1: Thank you for addressing previous comments. I have no reservations in recommending the revised manuscript for publication.

Reviewer #2: I believe that the authors addressed all my comments. There is, one point that should be considered - but I will not hold the paper on this - is still on the statistical analysis. The observed power relates to the observed effect size. The observed effect size (with insufficient sample) is an estimate and has large uncertainty around it. It could be that the "true" effect size is smaller. The issue appears when one considers the non-significant results that we cannot know that are truly below the expected efffect size or the analysis lacks power. I would suggest, for the future, that the authors calculate the power of the analysis also for the LMM.

The response on the Bayesian Statistics makes no sense still. The Bayes Factor can provide the odds for both alternative and the null. It makes no sense to assume the p-value threshold and then use a continuous outcome to qualify it (Bayes Factor) - these are separate approaches to inference. The fact that others used is no sound argument ("appeal to the people" or popularity fallacy).

7. PLOS authors have the option to publish the peer review history of their article (what does this mean?). If published, this will include your full peer review and any attached files.

Reviewer #1: **Yes: **James Mitchell

Reviewer #2: No

---

## [Editor Report · Acceptance letter]

2 Jun 2022

PONE-D-22-01772R1 

Visual control during climbing: Variability in practice fosters a proactive gaze pattern 

Dear Dr. Hacques:

I'm pleased to inform you that your manuscript has been deemed suitable for publication in PLOS ONE. Congratulations! Your manuscript is now with our production department. 

Kind regards, 

on behalf of

Dr. Greg Wood 

Academic Editor

PLOS ONE